



# Global VOC emissions quantified from inversion of TROPOMI spaceborne formaldehyde and glyoxal data

Yasmine Sfendla[1], Trissevgeni Stavrakou[1], Jean-François Müller[1], Glenn-Michael Oomen[1], Beata Opacka[1], Thomas Danckaert[1], Isabelle De Smedt[1], and Christophe Lerot[2]

[1]Royal Belgian Institute for Space Aeronomy (BIRA-IASB), Avenue Circulaire 3, Brussels, Belgium
[2]constellr, Brussels, Belgium (SA)

**Correspondence:** Trissevgeni Stavrakou (jenny@aeronomie.be)

**Abstract.** Volatile organic compounds (VOCs) are key precursors of tropospheric ozone and secondary organic aerosols, a major component of PM$_{2.5}$, and several aromatic VOCs are toxic. Glyoxal is a short-lived oxidation product of many VOCs, yet global models consistently underestimate its abundance, indicating a substantial missing source. Here, we derive improved estimates of global biogenic, pyrogenic, and anthropogenic VOC emissions and new constraints on the atmospheric glyoxal budget, based on the first joint inversion of TROPOMI formaldehyde and glyoxal columns using the adjoint of the MAGRITTEv1.2 chemical transport model. For 2021, the global NMVOC flux is estimated at 1070 Tg/yr, 19% above bottom-up estimates, partitioned into 749 Tg from vegetation, 102 Tg from biomass burning, and 219 Tg from anthropogenic activity. Emissions of anthropogenic glyoxal precursors are 43% higher globally when constrained by satellite data compared with inventory-based simulations, with large underestimations in India, China, and Africa. The total glyoxal source is estimated at 100 Tg/yr, of which 41% originates from unidentified VOCs, predominantly biogenic and concentrated in the Tropics. Likely contributors include poorly represented formation pathway in isoprene oxidation under low-NO$_x$ conditions and an underestimated contribution of monoterpenes. Validation against Pandonia Global Network, in situ, and MAX-DOAS datasets confirms improved agreement of the satellite-constrained model relative to the model based on inventory data alone.

## 1 Introduction

Volatile organic compounds (VOCs) are key precursors of tropospheric ozone, a harmful air pollutant and greenhouse gas (Chameides et al., 1988; Monks et al., 2015) and of secondary organic aerosols (SOA), a major component of PM$_{2.5}$ adversely affecting air quality and human health (Spracklen, 2011; Nault et al., 2021). In addition, some emitted VOCs themselves are toxic, carcinogenic, or cause respiratory irritation, especially aromatic hydrocarbons originating from anthropogenic activity such as benzene, toluene, ethylbenzene and xylenes (Lan et al., 2004; Partha et al., 2022). Thus, accurate quantification of nonmethane VOC (NMVOC) emissions is essential for tracking the effectiveness of clean air policies, and adresses a first major limitation in understanding and regulating the formation of smog and secondary air pollution: the uncertainty in the magnitude and distribution of surface NMVOC fluxes.





Emission inventories are generally the starting point when seeking a global picture of the VOC emission fluxes from anthropogenic activity, vegetation, and fires. Yet bottom-up inventories on their own are not sufficient, because they carry significant
uncertainties in activity data and emission factors, particularly in regions where field measurements or reporting of anthropogenic and pyrogenic activity are limited (Granier et al., 2011). Ground-based measurements alone do not suffice either because of their sparse and highly-localized nature. Large parts of the Tropics and the African continent, for example, are observational blind spots. Observation of directly-emitted NMVOCs from space remains limited due to the low concentrations and relatively weak spectral features of most VOCs in the spectral ranges accessible to satellite instruments. Nevertheless, sev-
eral key VOCs have been successfully retrieved, such as methanol (Razavi et al., 2011; Wells et al., 2014; Franco et al., 2024; Wells et al., 2025), acetone (Franco et al., 2019), formic acid (Razavi et al., 2011; Chaliyakunnel et al., 2016; Franco et al., 2018), isoprene (Fu et al., 2019; Wells et al., 2020), and ethene (Franco et al., 2022; Wells et al., 2025). While these retrievals have demonstrated global coverage, they often rely on assumed vertical profile shapes, exhibit limited sensitivity near the surface, require favorable thermal infrared conditions such as a strong surface-atmosphere thermal contrast, and have not yet been
integrated into inversion systems to constrain surface NMVOC fluxes: although these efforts provide valuable information for characterizing the large-scale distributions of the retrieved species and bottom-up emissions employed in large-scale models, they cannot readily offer constraints on the full range of different NMVOC species and sources. Secondary VOCs such as formaldehyde (HCHO) and glyoxal (CHOCHO), by contrast, are short-lived oxidation products of a wide range of NMVOCs and exhibit distinct spectral features in the UV and visible ranges, allowing for routine and reliable global retrievals (Thomas
et al., 1998; Wittrock et al., 2006). Although methane oxidation is the main source of formaldehyde at the global scale, and particularly over oceans (Stavrakou et al., 2009a), both formaldehyde and glyoxal over continental areas are predominantly produced by more reactive NMVOCs such as isoprene and monoterpenes over vegetated areas. Over heavily industrialized regions, anthropogenic precursors are often dominant; especially aromatic compounds and acetylene contribute substantially to the glyoxal budget in polluted areas (Fu et al., 2008; Stavrakou et al., 2009c). The two compounds are also directly emitted,
but in much smaller quantities than their secondary production. Formaldehyde is directly emitted by fuel combustion processes, biomass burning and vegetation. Glyoxal is released by fuel combustion (Qiu et al., 2020; Wang et al., 2023) and some industrial processes (Ho et al., 2013), and biomass burning (Zarzana et al., 2017, 2018).

Formaldehyde retrievals from space have been used to constrain VOC surface emissions (Pu, 2024; Oomen et al., 2024; Opacka et al., 2025) since the work of Palmer et al. (2003), while glyoxal has been used to that effect in combination with
formaldehyde (Cao et al., 2018; Chen et al., 2023) since the study by Stavrakou et al. (2009c). The short atmospheric lifetimes of these compounds (on the order of hours) and rapid formation from their parent NMVOCs make them effective tracers of recent VOC emissions: inverse modeling frameworks, such as the adjoint of the global MAGRITTE chemical transport model used in this work, use the satellite observations of these compounds together with detailed representations of atmospheric chemistry and transport to identify the NMVOC emissions that best explain the observed concentrations. In this work, emis-
sions from inventories are used as a priori estimates and subsequently optimized through the inversion of formaldehyde and glyoxal columns measured by the TROPOspheric Ozone Monitoring Instrument (TROPOMI) on board the Sentinel-5P (S5P)



satellite. While formaldehyde provides robust constraints on isoprene and the total VOC flux, glyoxal is more selective, for example to aromatic hydrocarbons, but has weaker signal strength.

A major limitation to our understanding of the impact of VOCs on atmospheric composition lies in the incomplete understanding of the atmospheric budget of some NMVOCs whose observed concentrations in the atmosphere are consistently much larger than model predictions. Two important examples of severely underestimated species are formic acid (Stavravkou et al., 2012; Millet et al., 2015) and glyoxal (Fu et al., 2008; Myriokefalitakis et al., 2008; Stavrakou et al., 2009c; Li et al., 2018; Silva et al., 2018; Lerot et al., 2023). A quantification of their atmospheric abundance and a better understanding of the distribution of their missing sources are the first steps towards an identification of the missing sources. The underestimation of continental glyoxal abundances by models has been attributed previously to a combination of factors, including underestimated emissions of known precursors (e.g. aromatics) and uncertainties in the glyoxal formation yields in the oxidation of NMVOC precursors such as isoprene (Li et al., 2016). In addition, glyoxal might be formed from the oxidation of NMVOCs that are currently not considered in models, such as furanoid compounds (Romanias et al., 2024). In this study, following Stavrakou et al. (2009c), the missing source of glyoxal is incorporated as the emission of an Unspecified Volatile Organic Compound (UVOC), representing uncharacterized, but presumably biogenic, organic compounds undergoing oxidation by the OH radical, ultimately yielding glyoxal. The assumption of biogenic origin is based on the dominance of biogenic sources in the global NMVOC budget and on the spatial correlation between observed glyoxal columns and forest coverage Stavrakou et al. (2009c). However, since forested regions are also frequent sites of biomass burning, a pyrogenic contribution cannot be excluded.

The use of satellite columns of formaldehyde and glyoxal together in a joint inversion directly addresses both knowledge gaps: formaldehyde provides strong constraints on total fluxes of reactive NMVOCs, while glyoxal offers sensitivity to specific emissions, namely those of glyoxal-producing precursors, including those not currently represented in emission inventories and atmospheric models. Some of the the limitations outlined above are addressed in this work by recent advances in chemical transport modeling and satellite observations. On the modelling side, the chemical mechanism has been updated to include a more accurate representation of aromatic hydrocarbon oxidation (Bates et al., 2021), which better reproduces results from chamber experiments. In addition, the parameterization of glyoxal uptake by aerosols has also been refined to account for relative humidity. On the observational side, the TROPOMI instrument offers daily global coverage at much finer spatial resolution and higher signal-to-noise ratio than its predecessors used in previous global model studies.

The manuscript is structured as follows. The observational datasets used in this work, namely the satellite data of formaldehyde and glyoxal acquired from TROPOMI as well as their validation, ground-based formaldehyde column data and in situ glyoxal concentrations, are presented in detail in Sect. 2. The simulation of formaldehyde and glyoxal with the MAGRITTEv1.2 chemistry-transport model and the design of the one-species and two-species inversions are described in Sect. 3. The top-down VOC emissions for different source categories and their evaluation through comparisons with independent observations are thoroughly discussed in Sect. 4. Conclusions are drawn in Sect. 5.



## 2 Observational datasets

### 2.1 TROPOMI formaldehyde and glyoxal column densities

TROPOMI was launched aboard the S5P satellite in October 2017. Functioning as an ultraviolet (UV), visible (VIS), near-infrared (NIR), and short wave infrared (SWIR) spectrometer with a spectral resolution of approximately 0.5 nm in the UV-VIS range, it operates in a low-Earth afternoon polar orbit with an equatorial overpass occurring daily around 13:30 mean local solar time. As of August 2019, it provides nearly daily global coverage at a spatial resolution of $3.5 \times 5.5 \ \mathrm{km}^2$ in the UV-VIS. It has a 2600 km swath divided into 450 ground pixels in the UV-VIS, and produces operational Level-2 (L2) products including vertical columns of $O_3$, $SO_2$, $NO_2$, HCHO, CHOCHO, CO, and $CH_4$, along with cloud and aerosol information (Veefkind et al., 2012).

The retrieval is based on a three-step differential optical absorption spectroscopy (DOAS) algorithm (De Smedt et al., 2018, 2021; Lerot et al., 2021). Firstly, as part of the calibration process, the absorption cross-sections are convolved with the instrumental slit function. Then the convolved cross-sections are fitted to the measured optical depths within a spectral window in the UV (for formaldehyde) or VIS range (for glyoxal) This step aims to determine the slant column densities (SCDs), representing the trace gas concentration integrated along the slanted effective light path through the atmosphere. The fitting procedure takes into consideration physical and instrumental effects to enhance the quality of the fit. To specifically target CHOCHO absorption bands, a fitting window spanning from 435 to 460 nm is employed, covering its two most intense bands (Barkley et al., 2017; Lerot et al., 2010). For HCHO, a fitting interval of 328.5 to 359 nm is used. The CHOCHO and HCHO absorption cross-sections are obtained from, respectively, Volkamer et al. (2005) and Meller and Moortgat (2000). For the glyoxal fit, Lerot et al. (2021) introduced two additional corrections to this step: one to minimize spectral misfitting caused by strong absorption of $NO_2$, and one to eliminate misfits caused by scene brightness inhomogeneities. In addition, a difficulty for the glyoxal fit lies in its spectral interference with water vapor, which can potentially lead to an overestimation of the glyoxal column density due to overlapping of their spectral features. This effect is particularly pronounced over oceanic regions where water vapor is abundant, and has also been observed in monsoon seasons over land (Chan Miller et al., 2014). Lerot et al. (2021) chose a water vapor cross-section at 293 K and 1013 hPa from the HITRAN2012 database.

Secondly, to convert the slant column densities to vertical column densities (VCDs, i.e., the concentration integrated from the Earth surface pixel beneath the satellite up to the top of the atmosphere), one requires the air mass factors (AMFs) in the middle of the fitting window (448 nm for CHOCHO, 340 nm for HCHO). These are obtained from a look-up table, which contains a range of precomputed altitude-resolved AMFs calculated with the radiative transfer model VLIDORT v2.6 (Spurr, 2008) for different combinations of observational parameters, such as observation angles, surface elevation, and surface albedo. From the table, the appropriate AMF based on the specific conditions of a TROPOMI observation is selected. The surface albedo is obtained from the Ozone Monitoring Instrument (OMI) minimum Lambertian-equivalent reflectivity (LER) climatology (Kleipool et al., 2008) at a spatial resolution of $0.5° \times 0.5°$ for CHOCHO as well as for HCHO at the respective wavelength of the AMF calculation. A priori vertical profiles at a $1° \times 1°$ resolution are provided by the global chemical transport MAGRITTEv1.1 (Müller et al., 2019) for CHOCHO, and TM5-MP (Williams et al., 2017) for HCHO. Over oceans,





the a priori CHOCHO profiles are obtained from airborne campaign measurements over the Pacific Ocean (Volkamer et al., 2015). Only observations with a cloud fraction less than 20% for CHOCHO and less than 40% for HCHO were retained for

processing, following the product recommendations.

Finally, background correction is necessary to mitigate systematic biases, such as instrumental effects or biases caused by spectral interference between the target absorber and other absorbers, and to correct for the presence of stripes in the SCDs derived from TROPOMI observations. To that end, observations in the Pacific Ocean, a remote and clean reference region, served as a baseline against which the observed CHOCHO and HCHO levels in other areas were compared and

130 corrected (De Smedt et al., 2021; Lerot et al., 2021). At the end of the procedure, a background column is added to the tropospheric column. For HCHO, this background value is taken from the model in the reference sector, ranging from 2 to $4 \times 10^{15}$ molec. cm$^{-2}$. For CHOCHO, a single offset of $0.1 \times 10^{15}$ molec. cm$^{-2}$ is added. Although some biases are removed in the DOAS-based algorithm (such as those caused by stripes, scene brightness inhomogeneities, or strong $NO_2$ absorption), it introduces systematic errors as well: SCD and spectral fitting uncertainties (e.g., absorption cross-section uncertainties and

135 interference with other species), AMF uncertainties (i.e., uncertainties of the input parameters in the AMF calculation and the profile shape) and background correction uncertainties (i.e., uncertainties of the reference VCD). Lerot et al. (2021) find that for CHOCHO VCDs, those three primary error components generally contribute nearly equally to the total systematic error. In low-emission remote regions with background-level CHOCHO VCDs ($< 0.2 \times 10^{15}$ molec/cm$^2$) however, spectral fit and background correction uncertainties dominate. The total systematic CHOCHO VCD errors were found to lie between 30% and

140 70% for regions with elevated (i.e., higher-than-background-level) CHOCHO emissions (Lerot et al., 2021). The systematic uncertainties for HCHO VCDs are reported in the Algorithm Theoretical Basis Document (ATBD) by De Smedt et al. (2018). For HCHO, the AMF uncertainty is the main contributor to the systematic uncertainty of the final product, both in regions with moderate HCHO VCDs ($> 5 \times 10^{15}$ molec/cm$^2$) and in regions with elevated columns ($> 8 \times 10^{15}$ molec/cm$^2$). The total systematic HCHO VCD retrieval error in regions with elevated columns (i.e., the Tropics and mid-latitudes in summer) was

145 estimated to be 35%, with the SCD and spectral fit uncertainties, AMF uncertainties, and background correction uncertainties contributing respectively 15%, 30%, and 10%. In regions with low VCD magnitudes, such as mid-latitudes in wintertime, the error can increase to 50–80%. While the retrieval uncertainty contains both a systematic and a random component, monthly averaging causes the random component to essentially vanish.

## 2.2 Satellite validation and bias correction

The TROPOMI CHOCHO data was compared with long-term multi-axis differential optical spectroscopy (MAX-DOAS) observations at a limited number of 8 sites in Asia and Europe and a ten-year near-continuous measurement record at Xianghe, China, by Lerot et al. (2021). They found a strong correlation (0.6–0.9) between the TROPOMI and ground-based CHOCHO columns for all stations except Bremen, Germany (0.1). A bias in the TROPOMI columns in wintertime could be responsible for the latter. While a strong correlation was found at the two most polluted sites (MAX-DOAS columns exceeding $5 \times$

$10^{14}$ molec. cm$^{-2}$) in Phimai, Thailand and Pantnagar, India, the mean biases between the TROPOMI and ground-based data were rather high, $-3.5 \times 10^{14}$ molec. cm$^{-2}$ in Pantnagar and $-2 \times 10^{14}$ molec. cm$^{-2}$ in Phimai. Significant negative biases are



not unusual for comparisons between satellite UV-VIS retrievals and MAX-DOAS data at strongly polluted sites (Verhoelst et al., 2021; De Smedt et al., 2021), and could be partly due to spatial heterogeneity and/or vertical sensitivity differences between the instruments. At the other six sites, biases did not exceed $\pm 0.9 \times 10^{14}$ molec. cm$^{-2}$, indicating an overall good agreement between the satellite and ground-based data.

The TROPOMI HCHO data (v.02.04.00) was characterized using the ground-based Fourier-transform infrared (FTIR) measurements from a larger number of 29 stations between May 2018 and November 2023 by Lambert et al. (2023) utilizing the method by Vigouroux et al. (2020). The FTIR vertical sensitivity profile is broadly similar to that of TROPOMI, with lower sensitivity closer to the surface, which makes FTIR especially suited for this validation. The median uncertainties of the FTIR observations were 13% (systematic) and $0.3 \times 10^{15}$ molec.cm$^{-2}$ (random) (Vigouroux et al., 2018). TROPOMI pixels were selected within 20 km of each FTIR site and comparisons with FTIR data obtained within a time window of $\pm 3$ hours of the satellite overpass time were included. Only pixels with a quality assurance value higher than 0.5 were used, which is the standard practice for TROPOMI products (De Smedt et al., 2023). This leads to about 30–40 TROPOMI-FTIR collocation pairs per site to average, but this number can be lower in cloudy conditions (Vigouroux et al., 2020). After minimizing vertical smoothing differences through the application of averaging kernels, a negative TROPOMI bias of $-30\%$ was found for high emission stations, a positive bias smaller than 20% for moderate-emission stations, and one of 32% for clean stations, in agreement with the reported systematic uncertainty of the product. With a robust Theil-Sen estimator, the slope and intercept of the TROPOMI HCHO columns as a function of the FTIR columns was determined, resulting in the following bias correction:

$$\Omega_{\mathrm{BC}} = 1.61 \times \Omega - 1.84 \times 10^{15} \text{ molec.cm}^{-2}, \tag{1}$$

with $\Omega_{\mathrm{BC}}$ the bias-corrected HCHO column and $\Omega$ the retrieved TROPOMI HCHO column. This bias correction is applied to the TROPOMI HCHO columns used in this work. A similar linear relationship was reported in the recent validation of OMI HCHO columns using FTIR column data and vertical profiles of in situ HCHO concentrations (Müller et al., 2024).

## 2.3 In situ glyoxal data

Field campaigns used for independent model evaluation in this study were carried out between 1988 (Grosjean et al., 1990) and 2018 (Liu, J. et al., 2020; Qian et al., 2019). A total of 70 in situ measurements were compiled from 37 sites across the globe, of which 17 were rural and 20 were urban locations. The distinction is not always clear-cut, such as for a site in Montelibretti, Italy (Possanzini et al., 2007), located in the countryside yet only 30 km away from Rome and therefore likely affected by urban pollution.

A variety of chemical and optical collection and analysis methods were employed in the compiled campaigns. Trapping of glyoxal on a solid sorbent and 2,4-dinitrophenylhydrazine derivatization followed by high-performance liquid chromatography (DNPH-HPLC) was used most commonly for the measurement of glyoxal concentrations. Historically, this chemical method has indeed been used since 1981 (Fung and Grosjean, 1981). It was used in the campaigns in references Lawson et al. (2015); Dai et al. (2012); Yang et al. (2018); Shen et al. (2018); Qian et al. (2019); Rao et al. (2016); Chang et al. (2019); Cerqueira et al. (2003); Possanzini et al. (2007); Borrego et al. (2000); Grosjean et al. (1996); Lee et al. (1995); Munger et al. (1995); Jing et al.



(2001); Grosjean et al. (2002) and Grosjean et al. (1990). In the work by Moortgat et al. (2002), Müller et al. (2005) and Ho
and Yu (2002), an O-pentafluorobenzyl hydroxylamine (PFBHA) coating was used instead of DNPH to collect the carbonyl
compounds. Liu et al. (2006) used dansylhydrazine (DNSH). Spaulding et al. (2003) implemented PFBHA derivatization
followed by gas chromatography-mass spectrometry (GCMS). Ieda et al. (2006) and Matsunaga et al. (2004) used an angular
denuder for collection, and gas chromatography for analysis. Optical methods used in this database include incoherent broad-
195 band cavity-enhanced absorption spectroscopy (IBB-CEAS) by Liu, J. et al. (2020); Min et al. (2016) and Washenfelder et al.
(2011), laser-induced phosphorescence (LIP) by Huisman et al. (2011); DiGangi et al. (2012) and Thayer et al. (2015), long-
path differential optical spectroscopy (LP-DOAS) by MacDonald et al. (2012) and Volkamer et al. (2005), and MAX-DOAS by
Hoque et al. (2018) and Sinreich et al. (2007). For MAX-DOAS measurements, column data were converted to volume mixing
ratios by the original authors Hoque et al. (2018); Sinreich et al. (2007) and used here as such. The altitude of in situ data
collection (i.e., except the MAX-DOAS measurements) varied between ground level and 90 m above ground level, depending
on the study.

In this work, the observed in situ CHOCHO concentrations are compared with the a priori and optimized concentrations
taking into account the location, altitude, time of the year and hour of the measurement. The campaign at the rural site Phimai
in Thailand (Hoque et al., 2018) differs from the others because it featured continuous measurements of both glyoxal and
205 formaldehyde from 2014 to 2016. Therefore, this data is used in an additional evaluation of the annual cycle of both com-
pounds. We acknowledge that the comparison with in situ campaigns has important limitations because the observations were
obtained for different years than the one studied here (2021). Additionally, due to the coarse model resolution ($2° \times 2.5°$)
representativeness issues in the data-model comparisons could be important, especially for urban locations. We therefore focus
our comparisons on rural locations, which are expected to be more representative of the modeled grid cell concentrations.

**2.4 Pandonia formaldehyde column data**

The Pandonia Global Network (PGN) (https://www.pandonia-global-network.org) is a network of real-time, standardized,
calibrated ground-based instruments measuring columnar trace gas concentrations of $NO_2$, HCHO, and $O_3$, using a passive
remote sensing spectrometer system capable of performing direct sun observations (Pandora). The main strengths of PGN are
the uniform instrument design and calibration, centralized data processing, data archiving, and distribution. The number of
215 Pandora instruments operating continuously is steadily growing. In this work, direct-sun HCHO column data available from
35 sites for the modeled year 2021 are used for model evaluation before and after the inversion (PGN, 2021). The total number
of PGN stations is higher, but HCHO is not yet provided at all stations. All sites with 2021 data are located in the Northern
Hemisphere, with the largest density in the eastern United States. More recently, sites in the Southern Hemisphere have been
progressively added to the network. We used stations providing data in HDF5 format, selecting only columns flagged as "high
quality (assured)" and "high quality (not assured)". Monthly averages of the observed columns were computed using days with
valid data.





## 3 Methodology

### 3.1 Formaldehyde and glyoxal simulated using the MAGRITTEv1.2 CTM

The Model of Atmospheric composition at Global and Regional scales using Inversion Techniques for Trace gas Emissions
(MAGRITTE) is a chemical transport model (CTM) capable of calculating the atmospheric distribution of 182 chemical com-
pounds. 141 of these undergo transport processes including advection, deep convection, and mixing within the boundary layer.
The model encompasses detailed up-to-date oxidation mechanisms for isoprene and other biogenic volatile organic compounds
(BVOCs) accounting for recent mechanistic advances (Müller et al., 2019). Photolysis rates within the model are obtained from
look-up tables calculated using the Tropospheric Ultraviolet and Visible radiative transfer model developed by Madronich and
Flocke (1998). MAGRITTEv1.1, and the new version described in this work, MAGRITTEv1.2, adopt most parameterizations
from the IMAGES model (Müller and Brasseur, 1995; Stavrakou et al., 2018), including the description of anthropogenic and
pyrogenic organic compound reactions (Stavrakou et al., 2009a; Bauwens et al., 2016). Calculations of chemical concentra-
tions are conducted over a $\sigma$-pressure coordinate system encompassing 40 vertical layers within the troposphere and the lower
stratosphere, extending up to a pressure level of 44 hPa.

All anthropogenic emissions are sourced from the Copernicus Atmosphere Monitoring Service (CAMS) CAMS-GLOB-
ANT inventory (Granier et al., 2019) and fire emissions from the QFED (Darmenov and da Silva, 2015) with emission factors
from Andreae (2019). Biogenic emissions are obtained from the MEGANv2.1 model embedded in the MOHYCAN canopy
environment model (MEGAN-MOHYCAN) (Stavrakou et al., 2018; Müller et al., 2008; Guenther et al., 2012; Bauwens
et al., 2018). The emissions are driven by meteorological fields from the ERA5 ECMWF meteorological reanalysis (Hersbach
et al., 2020). The spatial and temporal variability of the vegetation density is accounted for through the Leaf Area Index
(LAI) dataset obtained from MODIS Collection 6 reprocessed by Yuan et al. (2011). The effect of atmospheric $CO_2$ levels
on biogenic isoprene emission is accounted for based on the parameterization of Possell and Hewitt (2011). For $CO_2$ levels
of 416.4 ppm in 2021 (Lan et al., 2024), the $CO_2$ activity factor is equal to 0.90. While drought stress can also impact the
emissions, its effects are uncertain and therefore neglected in the model. Previous model evaluations against OMI data have
shown a deterioration of temporal correlation when accounting for the MEGANv2.1 soil moisture activity factor (Guenther et
al., 2006) and soil moisture fields from a meteorological reanalysis (Bauwens et al., 2018; Stavrakou et al., 2018; Opacka
et al., 2022).

In 2021, the global annual isoprene flux amounted to 433 Tg and the biogenic methanol emission flux (calculated following
Stavrakou et al. (2011)) to 137 Tg, available online at http://emissions.aeronomie.be. The global annual monoterpene flux
equaled 119 Tg. Biogenic emission of other compounds (ethanol, acetaldehyde, and acetone) are as described in Müller et al.
(2024).

Considering that aromatic compounds are important anthropogenic precursors of glyoxal, the CTM has been updated in this
work to better represent their oxidation through the integration of a new compact mechanism developed by Bates et al. (2021).
This oxidation mechanism offers the computational tractability of a relatively minimalist implementation of only 17 species
and 44 reactions, while providing a better match with observed yields from chamber experiments than more complex mecha-



nisms for hydrogen oxide radicals, glyoxal, and other oxygenates (Bates et al., 2021). The chemical mechanism of MAGRIT-TEv1.1 already included the oxidation of six glyoxal precursors: isoprene ($C_5H_8$), acetylene ($C_2H_2$), and ethene ($C_2H_4$), and the common BTX aromatics benzene ($C_6H_6$), toluene ($C_7H_8$) and xylenes ($C_8H_{10}$). The new mechanism in MAGRIT-TEv1.2 includes the following additional aromatics: trimethylbenzene ($C_9H_{12}$), styrene ($C_8H_8$), ethylbenzene ($C_8H_{10}$), phenol

($C_6H_6O$), cresol ($C_7H_8O$), catechols ($C_6H_6O_2$) and methylcatechols ($C_7H_8O_2$), benzaldehyde ($C_7H_6O$), (methyl-) perbenzoic acid ($C_7H_6O_3$), methylperoxybenzoylnitrate ($C_7H_5O_5N$), nitrophenols and nitrocatechols ($C_6H_5O_3N$), generic $C_4$ and $C_5$ intermediates ($C_4H_4O_2$ and $C_5H_6O_2$). Among these species, trimethylbenzene, styrene, and ethylbenzene are directly emitted by anthropogenic activities, and phenol, benzaldehyde, styrene, and ethylbenzene are released by fires.

A second update to the model is the revision of glyoxal reactive uptake by aerosols, following Curry et al. (2018). The

irreversible uptake of CHOCHO by aqueous aerosols was previously assumed to proceed with a uniform uptake probability, $\gamma_{RH} = 2.9 \times 10^{-3}$, based on laboratory experiments conducted under atmospheric conditions (Liggio et al., 2005). Evidence for rapid CHOCHO uptake is provided by direct measurements of gas-phase glyoxal (Volkamer et al., 2007; Ervens and Volkamer, 2010). We adopt the parameterization of the reactive uptake coefficients of CHOCHO by different cloud and aerosol types reported in Curry et al. (2018): the glyoxal uptake coefficient by sulfate, nitrate, or ammonium aerosols as a function of the

relative humidity (RH) is expressed as

$$\gamma_{RH} = \exp(12.1 - 44.5\ RH + 22.3\ RH^2). \tag{2}$$

This parameterization leads to much lower uptake at high RH ($\gamma_{RH} < 10^{-4}$ for RH > 80%) than in drier conditions (e.g., $\gamma_{RH} = 10^{-2}$ at RH = 50%). In the context of this work, it results in an overall decrease of the contribution of CHOCHO uptake to aqueous SOA formation compared to the use of the constant uptake probability $\gamma_{RH} = 2.9 \times 10^{-3}$. We note that

more research is needed to better understand the influence of other parameters, like temperature, aerosol acidity and organic content.

Table 1 summarizes the updated global annual of sources and sinks of atmospheric CHOCHO estimated in this study and in previous modeling studies. The overall a priori budget of CHOCHO (47.5 Tg yr$^{-1}$) remains consistent with previous estimates (Stavrakou et al., 2009c; Silva et al., 2018). The contribution of isoprene oxidation to this total is 23 Tg yr$^{-1}$, slightly less than

in the IMAGESv2 study (28 Tg yr$^{-1}$), due to mechanistic differences leading to a lower overall glyoxal yield from isoprene, as detailed in Müller et al. (2019). The larger photochemical production of glyoxal in this work (46 Tg yr$^{-1}$, as compared to 33 Tg yr$^{-1}$ in Silva et al. (2018)) results from several factors, including higher molar yields of glyoxal from aromatics in the mechanism of Bates et al. (2021) adopted in this work ($\sim$0.6 for BTX, as compared to 0.25 in Silva et al. (2018)), and higher glyoxal yields from monoterpenes (Müller et al., 2019).

The new parameterization of heterogeneous uptake does not cause an important change in comparison with the IMAGESv2 study, as the organic carbon and black carbon sink parameterization was left unchanged. However, the in-cloud CHOCHO sink has decreased significantly, from 4.7 Tg yr$^{-1}$ in the work of Stavrakou et al. (2009c) to 0.8 Tg yr$^{-1}$. The direct emission of glyoxal in this work and in the GEOS-Chem studies includes only pyrogenic emissions while Stavrakou et al. (2009c) included





**Table 1.** Global annual sources and sinks of atmospheric CHOCHO estimated in this work (average over 2021) by the a priori MAGRIT-TEv1.2 model, as compared to the a priori budget from Stavrakou et al. (2009c) for the year 2005, and the results of Silva et al. (2018) for the year 2005 as well. The sources and sinks are expressed in Tg of glyoxal per year. An aerosol uptake coefficient of $2.9 \times 10^{-3}$ was adopted in Fu et al. (2008), Stavrakou et al. (2009c) and Silva et al. (2018). Heterogeneous loss includes in-cloud uptake and aerosol uptake.

| CHOCHO budget | GEOS-Chem | IMAGESv2 | GEOS-Chem | MAGRITTEv1.2 |
| Tg yr$^{-1}$ | (Fu et al., 2008) | (Stavrakou et al., 2009c) | (Silva et al., 2018) | a priori (this work) |
|---|---|---|---|---|
| Direct emission | 8 | 9.0 | 6.5 | 1.5 |
| Photochemical production | 37 | 47.0 | 33.0 | 46.0 |
| **Total Source** | **45** | **56.0** | **39.4** | **47.5** |
| Photolysis | 28 | 28.7 | | 23.1 |
| OH oxidation | 6.5 | 10.0 | 35.7 | 9.1 |
| Heterogeneous loss | 6.4 | 12.4 | | 9.7 |
| Deposition | 4.1 | 4.5 | 3.7 | 5.6 |
| **Total sink** | **45** | **56.0** | **39.4** | **47.5** |
| **Global burden (Gg)** | 15 | 15.8 | 14.3 | 15.3 |
| **CHOCHO lifetime (hr)** | 2.9 | 2.5 | 3.2 | 2.8 |

a 3.2 Tg yr$^{-1}$ anthropogenic direct emission component. The direct pyrogenic source of glyoxal in this work (1.5 Tg yr$^{-1}$) is
lower than in the previous studies, due to the lower emission factors from Andreae (2019).

Finally, to facilitate a direct comparison with TROPOMI monthly averaged HCHO columns, the modeled monthly averaged columns are taken at the satellite overpass time ($\sim$13:30 local time), while accounting for the number of observations and averaging kernels provided with the TROPOMI retrievals. These kernels are applied to the modeled vertical profiles to account for the instrument's altitude-dependent sensitivity (Oomen et al., 2024).

**3.2  One-species and two-species inversion setup**

The MAGRITTE chemical transport model is used in a global adjoint-based inversion framework to optimize NMVOC emissions using TROPOMI observations. The emissions are iteratively refined through the minimization of a cost function $J$, which quantifies the discrepancy between modeled and observed columns, expressed as:

$$J(\mathbf{f}) = \frac{1}{2}\left(H(\mathbf{f}) - \mathbf{y}\right)^T \mathbf{E}^{-1}\left(H(\mathbf{f}) - \mathbf{y}\right) + \frac{1}{2}\mathbf{f}^T \mathbf{B}^{-1}\mathbf{f}, \tag{3}$$

with $\mathbf{f}$ denoting the vector of dimensionless emission parameters to be optimized, $H(\mathbf{f})$ the operation of the chemical transport model on the control variables, $\mathbf{y}$ the observation vector, $T$ indicating the transpose, $\mathbf{B}$ the covariance matrix of emission parameter errors, and $\mathbf{E}$ the covariance matrix of observation errors encompassing instrumental, representativeness, and model errors. The cost function is minimized using a quasi-Newton optimization algorithm, which involves calculating its gradient through the model's adjoint (Müller and Stavrakou, 2005). The convergence criterion is a reduction of the norm of the gradient
of the cost $J$ by a factor 30. Typically, this criterion is reached after approximately 30 iterations. The observation vector $\mathbf{y}$ and





**Table 2.** Global emission optimizations performed in this study for the target year 2021.

| Name | Description |
|---|---|
| OPTHCHO | One-compound inversion constrained by TROPOMI HCHO |
| OPTHCHOGLY | Two-compound inversion constrained by TROPOMI HCHO and CHOCHO, $\tau_{\mathrm{UVOC}} = 5\,\mathrm{days}$ |
| S1 | as OPTHCHOGLY, with $\tau_{\mathrm{UVOC}} = 1\,\mathrm{day}$ |
| S2 | as OPTHCHOGLY, with $\tau_{\mathrm{UVOC}} = 10\,\mathrm{days}$ |

its model counterpart $H(\mathbf{f})$ consist of monthly-averaged bias-corrected TROPOMI columns binned onto the model resolution ($2° \times 2.5°$). The optimized monthly-averaged top-down emission flux is expressed as

$$G(\mathbf{x}, t, \mathbf{f}) = \sum_{j=1}^{m} \exp(f_j) \phi_j(\mathbf{x}, t), \tag{4}$$

where $\phi_j(\mathbf{x}, t)$ indicates the a priori spatiotemporal emission distributions for each source category $j$ out of $m$ categories, and

$f_j(\mathbf{x}, t)$ are the emission parameters determined by the inversion for each category, model grid cell $\mathbf{x}$, and month $t$, in a given year.

The target year is 2021, with the simulation commencing on July 1st of 2020, exclusively incorporating continental data. Emissions are not optimized in grid cells for which the maximum monthly a priori emission (throughout one year) is very low, namely below $10^9$ molec. cm$^{-2}$ s$^{-1}$. Furthermore, only months with at least 10 valid satellite observations per model pixel are

included in the analysis.

### 3.2.1    Formaldehyde-constrained inversion

Table 2 summarizes the emission inversions conducted in this work. The formaldehyde-only inversion (hereafter abbreviated as OPTHCHO) optimizes three emission categories, namely biogenic, pyrogenic, and anthropogenic VOC fluxes (see Table 3). Biogenic fluxes are dominated by isoprene, but also include monoterpenes, ethanol, acetaldehyde and acetone. In the HCHO

inversion setup, around 111 000 emission parameters $f_j$ are inferred by the optimization: 42 000 (3500 grid cells $\times$ 12 months) for anthropogenic fluxes, 29 000 for biomass burning fluxes, and 40 000 for biogenic fluxes.

The observation covariance matrix $\mathbf{E}$ is assumed to be diagonal, with its diagonal elements the total (TROPOMI) observation uncertainties. Each one is calculated as the squared sum of the retrieval uncertainty, discussed in Section 2.1, and an absolute model uncertainty taken equal to $2 \times 10^{15}$ molec. cm$^{-2}$.

The emission covariance matrix $\mathbf{B}$'s diagonal elements are the squares of the relative errors, which are taken equal to $0.9$, i.e., the uncertainty factors of all emission parameters are assumed to be $e^{0.9} \approx 2.5$. The off-diagonal elements of $\mathbf{B}$ depend upon the spatio-temporal correlations of the errors in the fluxes. For biogenic and pyrogenic emissions, the spatial correlations are assumed to decrease exponentially between two grid cells, with the decorrelation length set to 300 km. Anthropogenic emission parameters within the same country are assumed to be weakly spatially correlated (coefficient of 0.1), while param-





eters for different countries are taken to be uncorrelated. The temporal correlation is assumed to be zero for biomass burning emissions. For biogenic emissions, it is assumed to decrease linearly from 0.6 for consecutive months to 0.1 after 6 months. For anthropogenic emissions, a similar linear decrease from 0.9 to 0.5 after 6 months is implemented.

**Table 3.** Emission categories optimized in the one-compound and two-compound inversions.

| Inversion constrained by HCHO | Inversion constrained by HCHO and CHOCHO |
|---|---|
| Biogenic | Biogenic |
| Biomass burning | Biomass burning |
| Anthropogenic VOCs (all) | Anthropogenic non-glyoxal precursors |
| | Anthropogenic glyoxal precursors (acetylene, ethene & aromatic hydrocarbons) |
| - | Missing glyoxal source |

### 3.2.2 Joint inversion constrained by formaldehyde and glyoxal columns

In addition to the source categories addressed by the one-species inversion described above, the simultaneous inversion of
HCHO and CHOCHO column data (hereafter abbreviated OPTHCHOGLY) also constrains the emissions of glyoxal precursors. By virtue of the adjoint-based inverse modeling approach, information gained from HCHO constrains the sources of CHOCHO, since both gases have common precursors (e.g., isoprene) and are interrelated through their chemical mechanisms (e.g., via OH). The focus here is limited to continental regions due to the inherent difficulty in retrieving CHOCHO columns over the oceans due to interference with liquid water absorption and because HCHO columns over oceans are mainly due to
methane oxidation (Stavrakou et al., 2009a).

In the joint inversion setup, besides the three emission categories of the formaldehyde-based inversion, two additional emission sources are optimized as summarized in Table 3 (i.e., $m = 5$). Firstly, the category of anthropogenic VOCs is subdivided into anthropogenic VOCs that are precursors of glyoxal (aromatic hydrocarbons, as well as acetylene and ethene) and anthropogenic VOCs that are not (ethane, propane, propene, formaldehyde, acetaldehyde, propionaldehyde, 2-butanone, formic acid,
acetic acid, butanes and higher alkanes, higher alkenes and alkynes). Secondly, a missing biogenic glyoxal source is introduced, presumed to form via photochemical oxidation (Stavrakou et al., 2009c) of unidentified VOC precursors (UVOC) with a molar yield of unity, resulting in a 5-day assumed lifetime. Additional inversions for the year 2021 with shorter (1 day) and longer (10 days) lifetimes are conducted as well, to assess the sensitivity of the model to the chosen UVOC lifetime. The a priori UVOC source is assumed to be distributed according to the MODIS leaf area index from Yuan et al. (2011) and scaled globally to
20 Tg yr$^{-1}$. The a priori anthropogenic CHOCHO precursors are described in Section 3.1. About 190 000 emission parameters $f_j$ are computed in the joint inversion for each year: 42 000 anthropogenic (besides glyoxal precursors), 29 000 pyrogenic, and 40 000 biogenic ones for the single-compound inversion, and an additional 42 000 for anthropogenic glyoxal precursors and 37 000 for the UVOC source.





For glyoxal, as for the formaldehyde columns, the elements of the diagonal observation covariance matrix $\mathbf{E}$ are the squared sum of the retrieval uncertainty, discussed in Section 2.1, and an absolute model uncertainty, which for glyoxal is taken equal to $1 \times 10^{13}$ molec. cm$^{-2}$.

The emission uncertainty factor in $\mathbf{B}$ is assumed to be $e^{0.9} \approx 2.5$ for the UVOC source, while for anthropogenic glyoxal precursors the factor depends on the geographical region. A factor of 2.5 is used for Canada, the United States, Oceania, Japan, and OECD (Organisation for Economic Co-operation and Development) member states in Europe. For all other regions, the uncertainty factor is taken to be $e^{1.1} \approx 3$. The decorrelation length of the UVOC source is set to 300 km, like for the biogenic and pyrogenic emissions, while its temporal error correlation is assumed to be constant at 0.3. Spatio-temporal correlations for the anthropogenic categories are identical to those in the single-compound inversion.

## 4   Results

### 4.1   Formaldehyde columns constrained by TROPOMI data

Overall, the HCHO columns from the a priori (bottom-up) model already agree very well with the bias-corrected satellite columns, both in terms of magnitude and spatial representation (Fig. 1). At regional scale, however, the a priori model often exhibits significant deviations from the observations. In the Northern Hemisphere, model underestimations occur during the local winter (Fig. 1a, c) in Southeast Asia, India and Mexico, and during the local summer (Fig. 1b, d) in Southern Europe, the Middle East, and the entire west of North America. In Central and Western Europe, the model overestimates HCHO columns. Over Siberia during summer, a good agreement is found, except at high latitudes ($>70°$), and the model successfully reproduces the location and magnitude of a major hot spot there that can be attributed to biomass burning (Fig. 2). In West and Central Africa, the a priori model agrees very well with the observations. In the Southern Hemisphere, in summer (Fig. 1a, c) overestimations are found in the semi-arid lowland Chaco Plain of South America and in the north of Australia, while underestimations are seen in Southern Africa. During the local winter, the model performs relatively well throughout the entire Southern Hemisphere (Fig. 1b, d).

After inversion (Fig. 1e, f), the aforementioned discrepancies with the a priori model are mostly eliminated, except for a few regions with low a priori emissions, such as the west of North America and the Middle East. For instance, year-round model underestimations of HCHO in India and in the moist savanna of Southern Africa (DRC, Angola and Zambia) are largely reduced, as well as the dry season mismatch in Southeast Asia.

### 4.2   VOC emissions inferred from bias-corrected TROPOMI formaldehyde columns

The annual top-down global emission fluxes over 2021 inferred from the OPTHCHO inversion are displayed in Fig. 2 and summarized in Table 4. The global fluxes of isoprene, biomass burning, and anthropogenic VOC emissions are moderately increased by the inversion, by 13%, 12% and 20%, respectively, compared to the bottom-up fluxes, but the inferred changes are significantly more pronounced at regional scale. The excellent top-down HCHO column agreement in Southern Hemisphere





**Figure 1.** HCHO columns averaged over December, January, February (DJF) and June, July, August (JJA) in 2021 ($10^{15}$ molec.cm$^{-2}$). (a, b): TROPOMI bias-corrected columns. (c, d): Columns from a priori model. (e, f): Columns optimized through the inversion constrained by TROPOMI HCHO columns (OPTHCHO).

Africa seen in Fig. 1 is attained thanks to the strong increase of emissions with respect to the bottom-up inventories used in the model across all categories: 85% more isoprene, 30 % more biomass burning VOCs and 40 % more anthropogenic VOCs (Table 4). In India, we find strong concomitant biogenic and anthropogenic flux increases with respect to the inventory (Fig. 2b, f) which are in good consistency with OMI-derived fluxes by Müller et al. (2024).





**Figure 2.** Emission fluxes (2021 average) from OPTHCHO study. Left panels show inventory-based global distributions of (a) isoprene, (c) biomass burning, and (e) anthropogenic VOC emissions used in the model. Blank pixels denote fluxes below $10^{10}$ molec. cm$^{-2}$ s$^{-1}$. The global a priori total flux per emission category is provided in each panel. Right panels show the ratio between the top-down (OPTHCHO-inversion) and bottom-up emissions of (b) isoprene, (d) biomass burning, and (f) anthropogenic compounds.

Significant increases of isoprene emissions as compared to the MEGAN-MOHYCAN-based model (Fig. 2a, b) are derived
in Europe and South Asia, by respectively 33% and 41%. In Oceania, the optimization induces a moderate decrease (by 25%).




**Table 4.** Bottom-up (a priori) and top-down continental emission estimates per source category for different world regions and globally, for the year 2021 (for the OPTHCHO and OPTHCHOGLY optimizations) or 2005-2017 (for the OMI 2005-2017 optimization by Müller et al. (2024)). Regions are defined in Supplementary Fig. S1. N.H.: Northern Hemisphere; S.H.: Southern Hemisphere. *In parentheses, global emissions are given for biogenic VOCs other than isoprene.

| | N. America | S. America | Europe | N.H. Africa | S.H. Africa | N. Asia | S. Asia | Oceania | **Global** |
|---|---|---|---|---|---|---|---|---|---|
| | | | | | Isoprene, Tg yr$^{-1}$ | | (monoterpenes, ethanol, acetaldehyde, acetone)* | | |
| MEGAN-MOHYCAN (a priori) | 35.2 | 142.2 | 8.7 | 94.1 | 45.9 | 11.4 | 39 | 56.6 | **433** (119, 22, 22, 28)* |
| OPTHCHO (this work) | 40.3 | 131.2 | 11.6 | 111.1 | 84.8 | 13.1 | 54.9 | 42.7 | **490** (140, 26, 26, 32)* |
| OPTHCHOGLY (this work) | 45.2 | 133.3 | 12.4 | 114.0 | 87.1 | 13.8 | 61.8 | 46.2 | **514** (147, 27, 27, 34)* |
| OMI 2005-2017 | 45.4 | 129.1 | 13.6 | 77.5 | 76.1 | 16.8 | 53.2 | 36.3 | **448** |
| | | | | | Biomass burning, Tg(VOC) yr$^{-1}$ | | | | |
| QFED (a priori) | 7.7 | 13.3 | 1.8 | 15.8 | 23.6 | 8.6 | 7.3 | 4.2 | **83** |
| OPTHCHO (this work) | 9.3 | 13.5 | 2.1 | 15.3 | 30.6 | 8.7 | 9.6 | 4.0 | **93** |
| OPTHCHOGLY (this work) | 9.0 | 14.2 | 2.0 | 18.0 | 34.9 | 8.8 | 11.2 | 4.2 | **102** |
| OMI 2005-2017 | 5.4 | 12.5 | 2.3 | 12.7 | 30.2 | 8.6 | 12.4 | 3.7 | **88** |
| | | | | | Anthropogenic VOC, Tg(VOC) yr$^{-1}$ | | | | |
| CAMS-GLOB-ANT (a priori) | 27 | 14.2 | 17.5 | 43.0 | 14.9 | 16.8 | 54.2 | 1.5 | **190** |
| OPTHCHO (this work) | 28.9 | 13.3 | 17.8 | 49.3 | 20.9 | 23.0 | 73.0 | 1.5 | **228** |
| OPTHCHOGLY (this work) | 27.4 | 13.3 | 17.6 | 48.0 | 19.0 | 23.3 | 68.3 | 1.6 | **219** |
| OMI 2005-2017 | 21.6 | 11.5 | 20.3 | 34.8 | 12.8 | 25.9 | 59.6 | 1.1 | **188** |
| | | | | *Anthr. CHOCHO precursors (included in Anthropogenic VOC)* | | | | | |
| *CAMS-GLOB-ANT (a priori)* | *3.3* | *1.9* | *3.6* | *4.0* | *1.5* | *3.6* | *17.4* | *0.3* | *35* |
| *OPTHCHOGLY (this work)* | *3.8* | *2.1* | *3.5* | *7.0* | *2.8* | *4.1* | *26.8* | *0.3* | *50* |
| | | | | | UVOC, Tg(VOC) yr$^{-1}$ | | | | |
| a priori | 2.6 | 5.8 | 1.5 | 2.0 | 2.0 | 2.1 | 3.6 | 0.7 | **20** |
| OPTHCHOGLY (this work) | 5.5 | 11.1 | 2.1 | 3.8 | 5.3 | 2.7 | 8.5 | 1.5 | **41** |

Smaller isoprene emission changes are derived throughout South America. The strongest local discrepancies, amounting to a factor of 2 to 3 locally between the bottom-up and optimized model, are observed in Turkey, India, Myanmar, Thailand, in the western part of North America (Canada, the U.S., and Mexico), in the Russian Far East, and in the moist savanna of Southern Africa (Angola, Zambia, Zimbabwe, Malawi, Tanzania). The pronounced increase in isoprene fluxes in Turkey are broadly

consistent with the study of Oomen et al. (2024), which relied on weekly bias-corrected TROPOMI HCHO over Europe, and inferred an emission increment of approximately a factor of 4. In comparison, less marked but positive emission increments (+50%) were derived based on OMI HCHO column data by Bauwens et al. (2016).

The TROPOMI-based OPTHCHO isoprene fluxes are compared with previous estimates based on bias-corrected OMI HCHO data (2005–2017, Müller et al. (2024)) in more detail in Fig. 3. Globally, the top-down isoprene estimates from these

two inversions are consistent in magnitude: 490 vs. 448 Tg yr$^{-1}$ for the TROPOMI and OMI inversions, respectively. The





spatial distributions of the isoprene emission ratios (Fig. 3b, d) are similar for the TROPOMI and OMI-constrained inversions in most regions that featured significant emission increases with respect to their bottom-up fluxes, such as in the western part of North America, Southern Hemisphere Africa, South Asia, and Turkey. Both inversions show little change, or even a decrease, in Northern Africa, the southern part of South America and Australia. Significant exceptions are Europe, North Asia and the Eastern United States, where the OMI-constrained inversion resulted in strongly enhanced isoprene emissions with respect to the a priori model, while no such enhancement is observed in the TROPOMI-constrained model. The reasons for these differences are unclear, but might be related to differences in the HCHO columns between OMI and TROPOMI, especially at mid-latitudes.

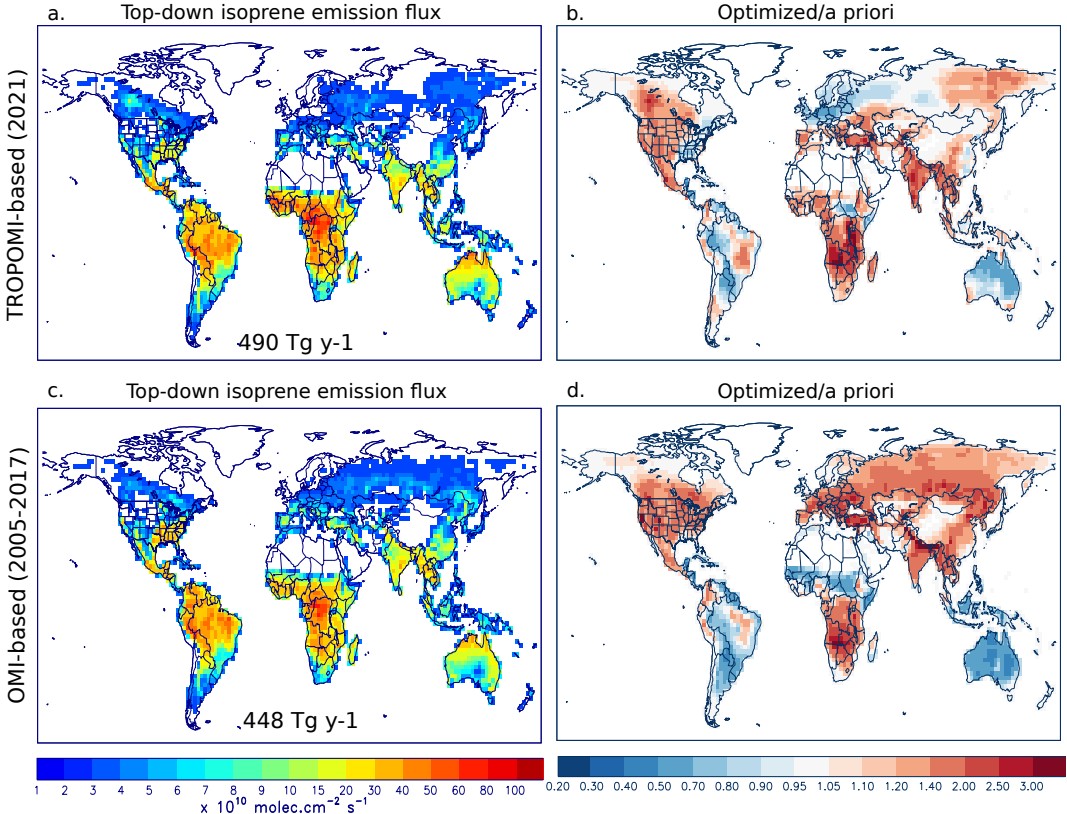

**Figure 3.** Modeled isoprene emission fluxes based on different satellite observations. (a): Average isoprene emission for 2021 constrained by TROPOMI formaldehyde columns, from this work (OPTHCHO inversion). (b): Average isoprene emission for 2005–2017 constrained by OMI formaldehyde columns, from Müller et al. (2024). Right panels show the ratio between the top-down and the bottom-up derived emissions of the (b) TROPOMI and (d) OMI inversion.

Significant differences between top-down and bottom-up (QFED) biomass burning emissions (Fig. 2c, d) are found in Southern Africa, especially over Zambia, Zimbabwe and Mozambique. This feature has been reported in previous inverse





modeling studies relying on SCIAMACHY and OMI HCHO data (Stavrakou et al., 2009b; Bauwens et al., 2016; Müller et al., 2024). Multiple other hotspots in Fig. 2d correlate with locations where agricultural fires are a common practice to prepare land for the upcoming planting season. For example, over north India, the large emission increase corresponds with annual post-monsoon crop residue burning in November (Lan et al., 2022). This results corroborates previous reporting of severe
inventory underestimations in the region (Liu, T. et al., 2020). Strong emission enhancements are also inferred in the North China Plain, where post-harvest burning is a common practice every year in May-June. These are in good agreement with previous estimates (Liu et al., 2015; Stavrakou et al., 2016; Lv et al., 2024). Likewise, the satellite data indicate a marked increase of fire emissions from European Russia and Belarus associated with human-induced fires in grasslands, arable, or abandoned lands occurring in spring and summer (McCarty et al., 2017) which are known to be underrepresented in bottom-up
inventories (Glushkov et al., 2021). Myanmar is a known biomass burning hotspot, where shrubland fires and agricultural fires constitute roughly half of the burnt area (Biswas et al., 2015). In March, pyrogenic emissions from the optimized model are twice as large as the bottom-up estimate there. Alongside those systematically occurring fires, episodic and sometimes extreme wildfires may occur in other regions, such as the boreal forests across Siberia (Ponomarev et al., 2023). A very large fire event took place in eastern Siberia in July–August 2021, which released over 3 Tg(VOC) according to QFED. It caused very large
TROPOMI HCHO columns (Fig. 1) which are well represented in the bottom-up model. The emission optimization indicates a moderate pyrogenic emission reduction, by 22% in these months, but its impact on modeled HCHO columns is more than compensated by a concomitant increase in isoprene fluxes in the same area (+29%). Intense forest fires also occurred across the northwestern United States in July 2021. In the states of Washington, Oregon, Idaho and Montana, the strong bottom-up model underestimation of the HCHO columns leads to almost a threefold increase of pyrogenic emissions: from 0.43 Tg of VOC
(July) in the bottom-up model to 1.2 Tg after optimization. In western Canada, in particular British Columbia and Manitoba, large emissions are reported by QFED (2.4 Tg VOC annually within 42–50°N and 90–125°W), but these are not significantly modified by the inversion (2.6 Tg) despite the large model underestimation of a priori HCHO columns in this region (Fig. 2). The improved (though still insufficient) representation of the formaldehyde columns there, after optimization, is primarily a result of enhanced biogenic emissions (+53% annually). Since the increase in biogenic emissions began a month earlier, in
June, well before the peak of the 2021 fire season (only ∼0.05 Tg of pyrogenic VOCs were reported in June), it is unlikely that source confounding occurred. One can therefore reliably conclude that biogenic summertime emissions, rather than pyrogenic ones, are underestimated by the emission inventories in Canada and the northwestern U.S.

Average anthropogenic VOC emissions (Fig. 2e, f) have nearly doubled in Iran and India. These changes are discussed in more detail in Section 4.4.

**4.3   Atmospheric sources and sinks of glyoxal**

The global top-down glyoxal budget is presented in Table 5. The global annual glyoxal source doubles from 48 Tg in the inventory-based model run (without UVOC) to 100 Tg after inversion. This result is in line with the SCIAMACHY-based inversion result for the year 2005 (Stavrakou et al., 2009c). For a small part (12 Tg), the source enhancement is due to increased emissions of known glyoxal precursors including isoprene (+19%), monoterpenes (+24%), pyrogenic VOCs (+23%)



and anthropogenic glyoxal precursors (+43%). The rest (41 Tg) is due to unidentified VOC precursors (UVOC). The top-
down emissions are discussed in more detail in Section 4.4. The glyoxal lifetime incurs little change through the optimization
(Table 5), similar to findings in Stavrakou et al. (2009c).

**Table 5.** Top-down (OPTHCHOGLY) global budget of atmospheric glyoxal (Tg yr$^{-1}$). UVOC: unidentified VOC precursors.

| **CHOCHO budget (Tg yr$^{-1}$)** | MAGRITTEv1.2 Top-down (this work) | |
| --- | --- | --- |
| Direct emission | 1.9 | |
| Photochemical production | 98.4 | |
| *from known sources* | | *57.7* |
| *from UVOC* | | *40.7* |
| **Total Source** | **100.3** | |
| Photolysis | 54.4 | |
| OH oxidation | 18.9 | |
| Heterogeneous loss | 17.1 | |
| Deposition | 9.9 | |
| **Total sink** | **100.3** | |
| **Global burden (Gg)** | **32.9** | |
| **CHOCHO lifetime (h)** | **2.9** | |

The glyoxal columns from the two-compound OPTHCHOGLY inversion are displayed in Fig. 4 and Fig. 5. TROPOMI mea-
sures high year-round glyoxal column abundances in the Tropics. In the northern and southern temperate zones, the columns
peak during local summer. Although the a priori model reproduces relatively well the patterns of the observed glyoxal distri-
bution (Fig. 5a–d), the magnitudes are strongly underestimated. The gap is largely closed after inversion (Fig. 5e, f).

After inversion, the glyoxal columns remain underestimated in many arid and semi-arid areas, including the Arabian Penin-
sula (by about a factor of 3) and, during summer, northwestern China and Inner Mongolia (locally by more than a factor of 5),
and the Western United States (by a factor of 1.5), i. e. low-emission areas according to emission inventories. In the Western
U.S., formaldehyde columns remained underestimated after inversion as well (see Section 4.1). In Northwestern China and
Inner Mongolia, the mining, combustion and processing of coal have intensified drastically since 2010, and the accuracy of the
related emissions in bottom-up inventories is relatively low (Zhu et al., 2022). Similarly, it has been found in numerous studies
that anthropogenic emission inventories do not accurately represent observations around the Red Sea and the broader Middle
East region (Osipov et al., 2022). These factors likely contribute to the lower model performance in those regions.
Fig. 5b and d show a severe underestimation of a priori glyoxal columns in summer in the extra-tropical Northern Hemi-
sphere, especially over the global boreal zone and over broadleaf and mixed zones across the U.S. and Europe. This discrepancy
is corrected after inversion (Fig. 5f, Fig. 4b, f). The effect can be seen in more detail for the broadleaf zones in the time series
in Fig. 4b and f: in the Southeast U.S. during summer, the a priori columns are half the observed values, and the difference





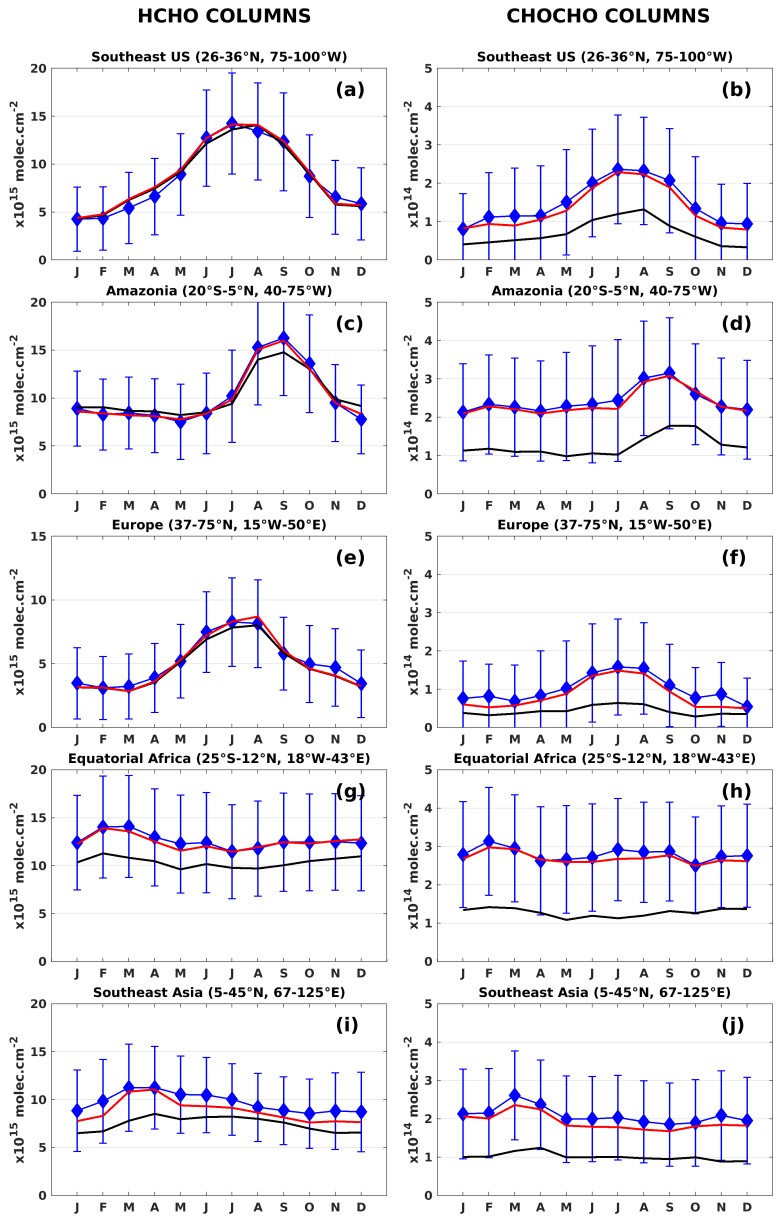

**Figure 4.** Monthly mean formaldehyde (left) and glyoxal (right) columns simulated with MAGRITTEv1.2 and observed by TROPOMI (blue diamonds) for 2021. The regions are defined as bounding boxes with coordinates given in each subfigure and shown on Supplementary Fig. S1. Error bars correspond to the errors used in the inversion (cf. Section 3.2). Black solid lines correspond to the a priori simulation, red to the optimized columns of the OPTHCHOGLY inversion. The optimized HCHO columns from the OPTHCHO inversion (not shown) are very similar to the OPTHCHOGLY results. Units are $10^{15}$ molec. cm$^{-2}$ for HCHO, $10^{14}$ molec. cm$^{-2}$ for CHOCHO.





**Figure 5.** CHOCHO columns averaged over December, January, February (DJF) and June, July, August (JJA) in 2021 ($10^{14}$ molec.cm$^{-2}$). (a, b): TROPOMI columns. (c, d): Model columns with a priori (bottom-up) emissions, without the UVOC a priori emissions. (e, f): Model columns optimized through the top-down inversion constrained by TROPOMI HCHO and CHOCHO columns (OPTHCHOGLY).

vanishes after optimization (Fig. 4b). Over Europe, the a priori model discrepancy is smaller in winter, while in summer, the
measured column exceeds the a priori values by about 150%, with the gap closing after inversion (Fig. 4f).

In tropical regions, the observed seasonality is relatively weak and is well captured by the model. The peak of glyoxal concentration around September results from a combination of direct emissions from fires and photochemical production from





pyrogenic hydrocarbons (Kluge et al., 2023), as well as biogenic emissions, which are more pronounced in the dry season. The bottom-up underestimation of the glyoxal columns by a factor of 2 year-round gives way to an excellent match in the top-down

model (Fig. 4).

## 4.4    Top-down VOC emissions inferred using TROPOMI formaldehyde and glyoxal columns

As compared to the formaldehyde-constrained inversion (OPTHCHO), the additional constraints provided by glyoxal observations (OPTHCHOGLY) increase the pyrogenic emissions by 10% globally, whereas isoprene and anthropogenic emissions show small changes (within 5%) (Table 4). Most significantly affected are the pyrogenic emissions in Africa and South Asia,

where the total annual emissions are increased by circa 15% in OPTHCHOGLY relative to OPTHCHO inversion. The only significant change in isoprene emissions occurs in North America and South Asia, which both increase by circa 12% compared to the OPTHCHO model. For most other regions, the optimized OPTHCHOGLY emissions remain close to the OPTHCHO results (Table 4 and Fig. 6). Although, for each of the three categories, the global total annual emissions derived in this work (either OPTHCHO or OPTHCHOGLY) are higher than both the bottom-up estimates and the estimates by Müller et al. (2024),

regionally, the top-down isoprene emissions derived from OMI columns by Müller et al. (2024) are higher than those obtained here at northern extratropical latitudes (North America, Europe and North Asia), as well as the anthropogenic VOC emissions in Europe and North Asia (Table 4). Those discrepancies are likely due to differences between (bias-corrected) OMI and TROPOMI columns.

In every category, Indian emissions are drastically larger than in the inventories: yearly isoprene emissions are more than

doubled (from 8 to 17 Tg), pyrogenic emissions are tripled (to 3 Tg), and anthropogenic VOCs are 1.6 times larger (from 12 to 19 Tg). The latter is more in line with other recent assessments in the literature, e.g. 16 Tg for the year 2015 according to the bottom-up inventory of Venkataraman et al. (2020) and 20 Tg in 2009 for the entire indian subcontinent in the OMI-based top-down estimation by Chaliyakunnel et al. (2019). Due to rapid economic growth and limited regulations targeting anthropogenic VOC sources in India (Ganguly et al., 2020), these emissions show a significant positive trend based on long-term OMI HCHO

data (Bauwens et al., 2022; Müller et al., 2024) and are expected to have increased steadily up to 2021.

Two additional categories are optimized in the OPTHCHOGLY setup: the anthropogenic glyoxal precursors (Section 4.4.1) and the unspecified glyoxal precursor UVOC (Section 4.4.2). The former includes all main aromatic compounds (benzene, toluene, ethylbenzene, xylenes, trimethylbenzene, and styrene), as well as acetylene and ethene. All categories included, the annual global VOC source is estimated in the OPTHCHOGLY study at 1070 Tg, up by 19% from the bottom-up estimation of

897 Tg.

### 4.4.1    Top-down emissions of anthropogenic glyoxal precursors

According to the CAMS-GLOB-ANT inventory, the main source regions of anthropogenic glyoxal precursors are the eastern United States, Central Europe, Southeast Asia, and most significantly India and eastern China (Fig. 6j, Supplementary Fig. S2). The top-down inversion (Fig. 6k and 6l) results in a threefold emission increase in India, as well as moderate increases in

Mexico, the Middle East, Indonesia, and northern China. In a few areas in the Middle East and in northwestern China, much



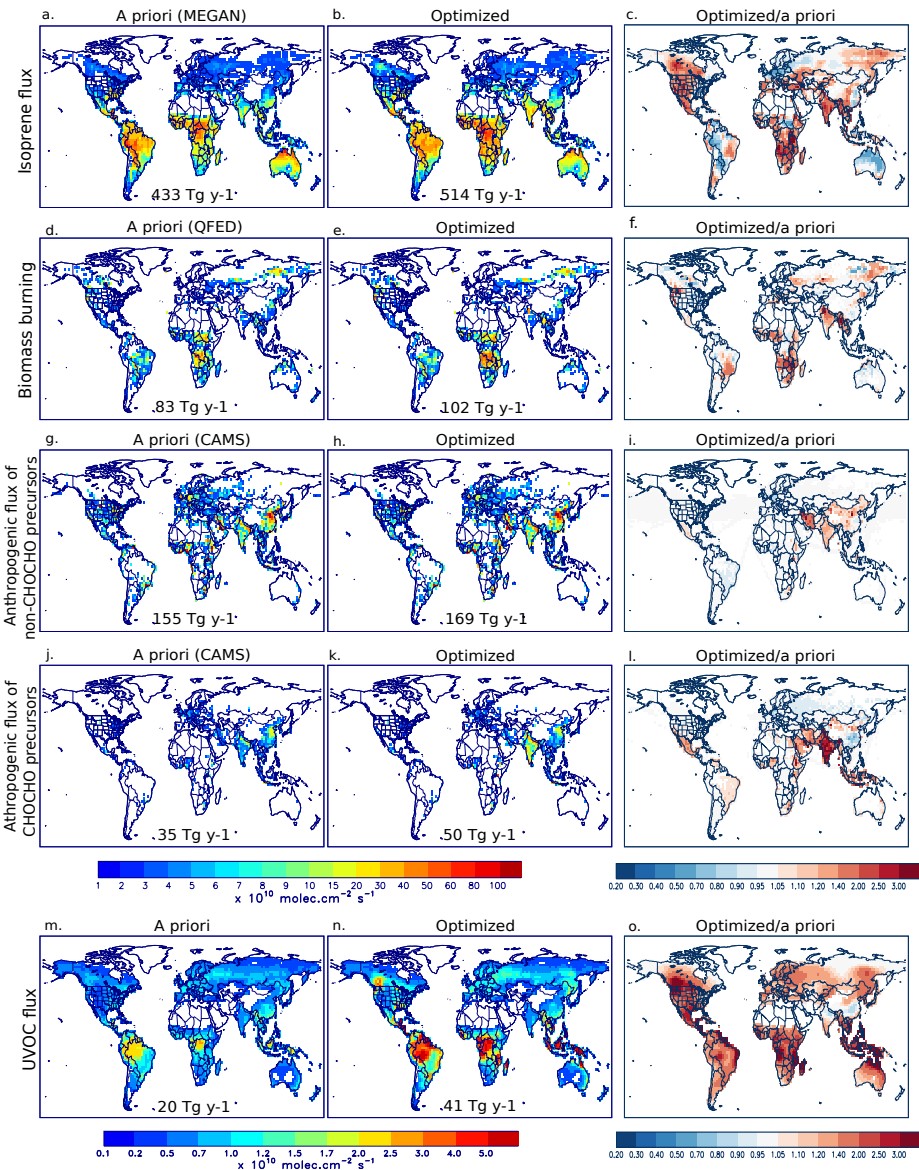

**Figure 6.** Global distributions of bottom-up emissions (left) and top-down emissions (middle) from the OPTHCHOGLY inversion over 2021, for isoprene (a, b), biomass burning VOCs (d, e), anthropogenic VOCs non-glyoxal precursors (g, h), anthropogenic glyoxal precursors (j, k), and the UVOC glyoxal precursor (m, n). The global total flux per emission category for 2021 is given inset. Right panels show the ratio of the top-down by the a priori emissions.



higher local increases (factors of 1.5–3) are derived, even though the optimized glyoxal columns remain low compared to the observations (Fig. 5). Cumulatively across Africa, anthropogenic glyoxal precursors are emitted at nearly twice the amount (9.8 Tg yr$^{-1}$) reported in the inventory (5.5 Tg yr$^{-1}$) (see Table 4 and Fig. 6l). Slight decreases with respect to the inventory are found in North America, North Asia, and Central and South China. Globally, the emissions of anthropogenic glyoxal

precursors are estimated at 50 Tg yr$^{-1}$ after inversion, an increase of 43% relative to the bottom-up estimate (see Table 4).

According to the OPTHCHOGLY inversion, China emitted 6.5 Tg of anthropogenic aromatics in 2021, very close to the bottom-up estimate (Table 6). The distribution, however, differs significantly between the bottom-up and top-down model. The OPTHCHOGLY emission estimates exceed the a priori fluxes in the Beijing–Tianjin–Hebei (BTH) and Yangtze River Delta urban areas but are lower in the southern urban clusters including Wuhan, Chongqing and the Pearl River Delta (Fig. 7c).

The derived source of aromatics from China is about twice lower than reported in the satellite-constrained work of Liu et al. (2012a) for 2007 (13.4 Tg yr$^{-1}$). This disparity can be attributed to differences in the satellite retrievals of SCIAMACHY and TROPOMI and possibly to a decrease in the emissions of glyoxal precursors between 2007 and 2021: the SCIAMACHY glyoxal columns reported by Liu et al. (2012a) were in the range (5–10)×10$^{14}$ molec. cm$^{-2}$ in August 2007 over Eastern China, which is about twice higher than the TROPOMI columns in August 2021 in the same region. Spaceborne glyoxal data from

multiple missions, including OMI, GOME-2 and TROPOMI indicated slightly lower columns since 2015, compared to previous years, possibly in response to emission regulations (Lerot et al., 2021). This decrease (of the order of 10$^{14}$ molec. cm$^{-2}$) does not completely explain the very high values used in Liu et al. (2012a), which are therefore likely due to retrieval differences.

The Beijing-Tianjin-Hebei region (BTH) implemented numerous programs to address air pollution since 2012 (Xiao et al., 2020). Our findings suggest that BTH emits about 30% more aromatics than in the bottom-up inventory (Table 6). The relative

contribution of aromatics to the total anthropogenic VOC emissions in BTH is similar in the bottom-up model and after inversion (19%), but lower than in the bottom-up estimate by Li et al. (2019) for 2015 (33%). Ambient measurements near Beijing revealed that the aromatic emission flux was significantly lower than reported in Li et al. (2019), suggesting that the contribution of aromatics should be lower (∼10%), i.e. closer to the top-down estimate in this work. A partial explanation could be governmental regulations in the region (Simayi et al., 2022). In view of the disparities among studies (Choi et al.,

2024), it is clear that more detailed investigations are needed.

The total of anthropogenic VOC and aromatic emissions over the Yangtze River Delta (YRD) region changed little after inversion (Table 6). Both the bottom-up and top-down estimates agree well with those from the bottom-up inventory of An et al. (2021), based on local measurements for 2017. In terms of seasonal variation, the top-down aromatic fluxes in Shanghai peak in July, a feature absent from the inventory (Supplementary Fig. S3). This result is consistent with the strong temporal

variation of aromatic fluxes in Shanghai, reported by Wang et al. (2020), with maxima in winter (December/January) and summer (June/July). We observed this seasonal feature to a lesser extent in Guangzhou, but not in Beijing. The consistent winter peaks in the CAMS-GLOB-ANT inventory likely reflect heating-related emissions that are relatively well represented, whereas summertime activities may be underrepresented in the inventories. Wang et al. (2020) suggest that seasonal activities (such as outdoor painting) and the influence of meteorology on evaporation processes are key factors contributing to these

summer peaks of aromatic emissions missed in bottom-up inventories.



**Table 6.** Anthropogenic VOC and aromatic hydrocarbon emissions in China, BTH (Beijing-Tianjin-Hebei) and YRD (Yangtze River Delta). BU: bottom-up; TD: top-down; ant. VOCs: anthropogenic VOCs; % aromatics: percentage of aromatic hydrocarbon emissions with respect to total anthropogenic VOC emissions.

| Study | Method | Year | Ant. VOCs | Aromatics | % Arom. |
|---|---|---|---|---|---|
| | | | **Average emissions (Tg yr$^{-1}$)** | | |
| | | | China | | |
| Liu et al. (2012a), a priori | BU (Zhang 2009) | 2007 | 23 | 2.4 | 10 |
| Liu et al. (2012a) | TD (SCIAMACHY CHOCHO) | 2007 | 34 | 13.4 | 39 |
| This work, a priori | BU (CAMS-GLOB-ANT) | 2021 | 31 | 6.4 | 21 |
| This work, OPTHCHOGLY | TD (TROPOMI HCHO & CHOCHO) | 2021 | 37 | 6.5 | 18 |
| | | | Beijing-Tianjin-Hebei | | |
| Li et al. (2019) | BU | 2015 | 3.3 | 1.1 | 33 |
| This work, a priori | BU (CAMS-GLOB-ANT) | 2021 | 7.4 | 1.4 | 19 |
| This work, OPTHCHOGLY | TD (TROPOMI HCHO & CHOCHO) | 2021 | 9.5 | 1.8 | 19 |
| | | | Yangtze River Delta | | |
| An et al. (2021) | BU (local observation-based) | 2017 | 4.9 | 1.2 | 25 |
| This work, a priori | BU (CAMS-GLOB-ANT) | 2021 | 6.4 | 1.4 | 22 |
| This work, OPTHCHOGLY | TD (TROPOMI HCHO & CHOCHO) | 2021 | 5.9 | 1.3 | 22 |

In the northwestern Xinjiang region of China, emission hotspots are found around Ürümqi, as well as a particularly high flux 500 km southwest from that city, in a sparsely populated area. There, the optimization indicates that aromatic emissions are a factor of 4 higher than the bottom-up estimate. These emissions are likely released from the large cotton textile manufacturing industry and from raw textile dyeing wastewater (Ning et al., 2015). During the studied year, about 90% of China's cotton was 540 grown in Xinjiang, or about a quarter of the world's total (Gale and Davis, 2022). Elsewhere in China, the top-down model suggests anthropogenic emissions of glyoxal precursors similar to or slightly lower than the inventory, e. g. in the Pearl River Delta (PRD). The latter reflects a clear evolution since 2006–2007, when Chan Miller et al. (2016) found a very intense glyoxal hotspot over the PRD observed by OMI, much higher compared to the rest of the industrial coast.

In India, the top-down model suggests annual top-down emissions of anthropogenic glyoxal precursors nearly three times 545 higher than those in the CAMS-GLOB-ANT inventory (8.2 Tg vs. 2.9 Tg). From the top-down perspective, the annual total emissions of anthropogenic glyoxal precursors from India and China (8.1 Tg) are similar, whereas China is globally dominating the emission of anthropogenic aromatics (6.5 vs. 3.7 Tg from India). The dominance of aromatic emissions from China follows from the VOC speciation of the inventory (Supplementary Fig. S2 and Fig. 7). In India, we see that the very large discrepancies between the top-down and inventory emissions of glyoxal precursors are likely mainly caused by an underesti-550 mation of acetylene and ethene emissions, and to a lesser extent of aromatics. The threefold flux increase from India explains



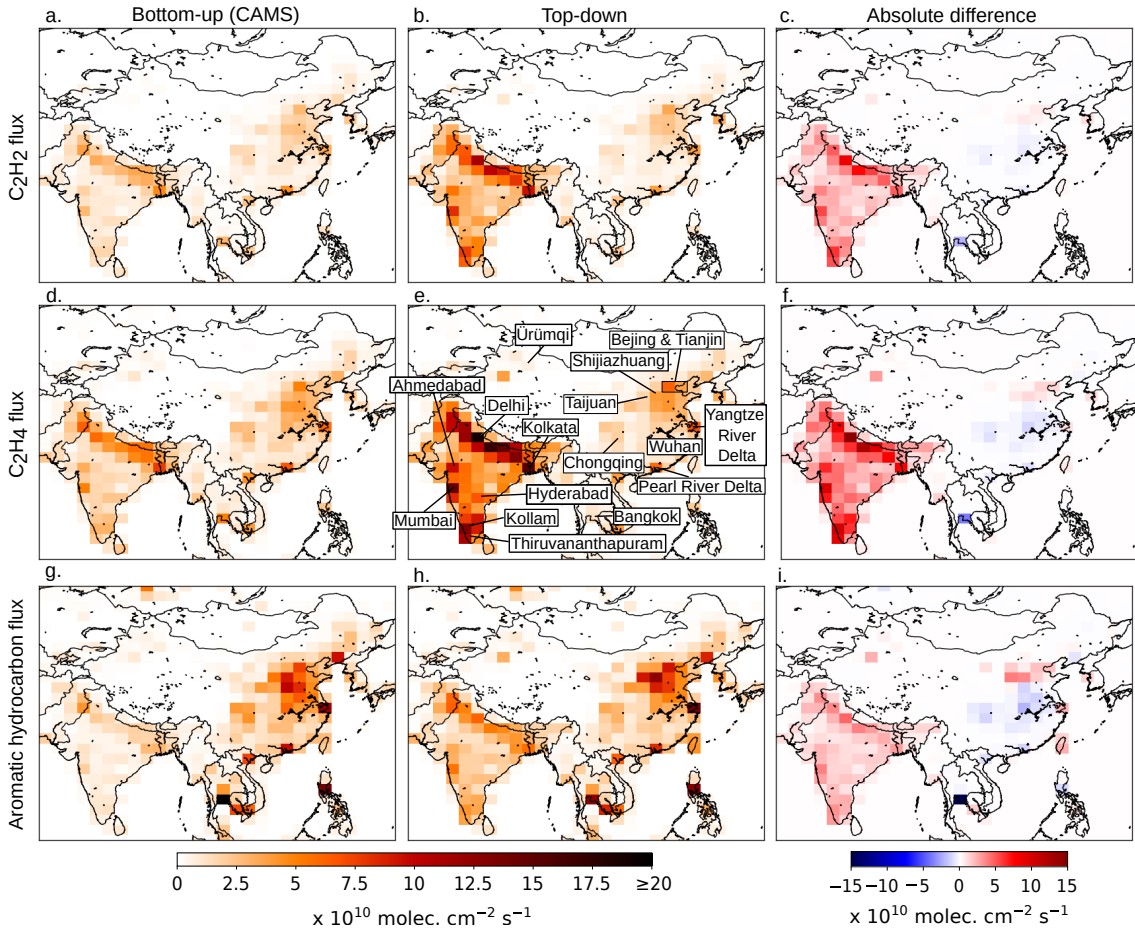

**Figure 7.** Distributions of anthropogenic bottom-up (a, d, g) and top-down (b, e, h) emissions of acetylene, ethene and aromatic hydrocarbons in India, China and South Asia averaged over 2021. The bottom-up emissions are based on CAMS inventory data. The top-down emissions are derived by the OPTHCHOGLY inversion constrained by HCHO and CHOCHO data. The absolute difference between the top-down and bottom-up model is given in c, f, and i.

the substantial enhancement of global anthropogenic emissions of acetylene and ethene at the global scale (+57%), whereas the global aromatic emission changes are more moderate, of the order of 20% (Table 7).

The discrepancies between bottom-up and top-down emissions might have several causes. The analysis of spaceborne infrared observations of ethylene suggests that the large point sources of this compound are strongly underestimated in bottom-up inventories (Franco et al., 2022), in particular over India, northern China, the Middle East and Europe. These flaws likely concern also other VOCs, such as acetylene and aromatics. Over India, the underestimation of bottom-up emissions might be also related to the large share of domestic emissions and informal, small-scale industries in this country (Mukim, 2015). In





**Table 7.** A priori (bottom-up) and optimized (top-down) global anthropogenic emissions of glyoxal precursors derived from the OPTHCHO-GLY inversion for 2021. Aromatic hydrocarbons are italicized. Emissions are expressed in Tg yr$^{-1}$. The percentage emission increase for each precursor is in parentheses.

| Anthropogenic glyoxal precursors | A priori | Top-down (this work) |
|---|---|---|
| acetylene ($C_2H_2$) | 2.80 | 4.37 (+56%) |
| ethene ($C_2H_4$) | 6.48 | 10.18 (+57%) |
| *benzene* ($C_6H_6$) | 5.65 | 7.88 (+39%) |
| *toluene* ($C_7H_8$) | 7.76 | 9.06 (+17%) |
| *xylene* ($C_8H_{10}$) | 7.36 | 8.40 (+14%) |
| *trimethylbenzene* ($C_9H_{12}$) | 0.91 | 1.11 (+22%) |
| *Other aromatics (styrene, ethylbenzene)* | 4.48 | 5.73 (+28%) |

comparison, Chinese industries tend to be state- or privately owned large-scale enterprises, which are more tightly regulated and better represented in inventory activity data. Additionally, inventories are limited by the lack of activity data and emission

factors specific to Indian emission sources (Stewart et al., 2021). Emission sources particular to India include industrial sources such as brick kilns and the processing of textiles, jute and leather; and domestic ones such as residential cooking, home use of poorly-serviced diesel generators, and burning of solid fuel and plastic waste for heating. This is further complicated by the ubiquity of inferior-quality coal, diesel and petrol, which impacts the chemical profile of traffic, residential and industrial emissions (Kumar et al., 2016; Stewart et al., 2021). Hence, emission factors determined for a similar activity in a different

country might not translate well to the Indian context.

### 4.4.2    Unspecified glyoxal precursor emissions

The TROPOMI-constrained analysis indicates a total global annual UVOC source of 41 Tg (Table 4). In comparison, a similar inversion study constrained by SCIAMACHY data by Stavrakou et al. (2009c) for 2005 derived a higher UVOC source after optimization (54 Tg yr$^{-1}$), even though the total glyoxal source they obtained was similar to the one here, of 100 Tg.

The sensitivity of the model to the UVOC lifetime is shown in Fig. 8. The inferred distribution of the UVOC source is similar across the inversions, and the global totals are respectively 34, 41 and 47 Tg yr$^{-1}$ when the UVOC lifetime is set to 1, 5 and 10 days. Longer UVOC lifetimes allow emitted UVOCs to move away further from the location where they were emitted (always on land) before being oxidized into glyoxal (over land and ocean). It is therefore expected that increasing the UVOC lifetime decreases the glyoxal production over land, thereby requiring a larger continental UVOC emission flux to close the gap

between the modeled and TROPOMI-observed glyoxal columns. The sensitivity analysis is discussed further in Section 4.6.

We find that circa 70% of the entire UVOC emission flux stems from the Tropics. The largest increases relative to the assumed a priori source of 20 Tg are also derived over the Tropics, especially in South Asia (+136%) and Southern Hemisphere Africa (+165%). Substantial increases are also found in Northern Hemisphere Africa and South America (+90%), and the UVOC source nearly doubles in North America, primarily due to large increases over Mexico, Central America, the northwestern



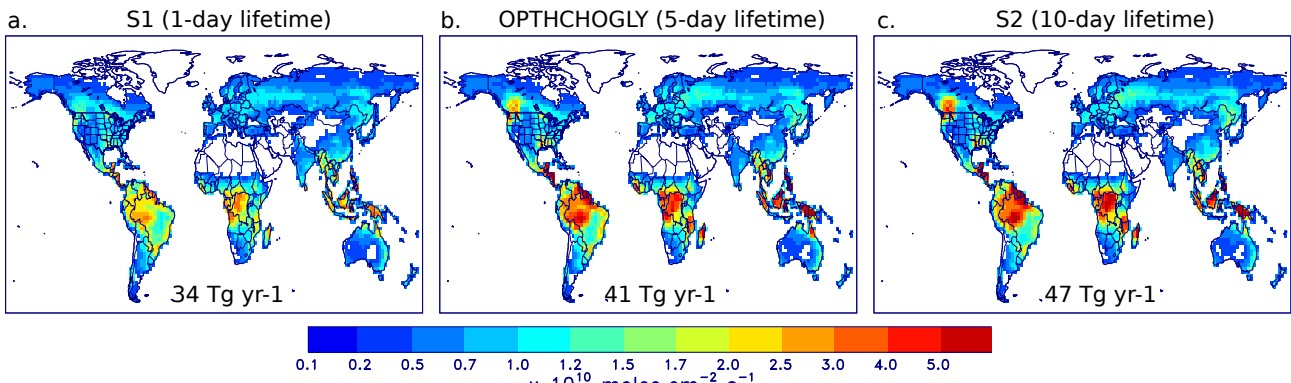

**Figure 8.** Global annual distribution of the top-down source of the unspecified glyoxal precursor (UVOC) (a) for an assumed UVOC lifetime of 1 day (inversion S1), (b) 5 days (OPTHCHOGLY inversion) and (c) 10 days (S2). The global annual flux is provided in each panel.

United States, and western Canada (Fig. 6o). This suggests a substantial underestimation of glyoxal production from biogenic precursors, likely reflecting a combination of underestimated VOC fluxes and incomplete representation of their oxidation chemistry in the model.

In the boreal zone, where monoterpenes dominate BVOC emissions (Guenther et al., 2012), the a priori underestimation of glyoxal (Section 4.3) is likely linked to monoterpenes, whose emissions may be too low in the MEGAN-MOHYCAN inventory 585 and/or whose glyoxal yields are underestimated in the model, consistent with previous findings based on OMI glyoxal data and the GEOS-Chem model (Silva et al., 2018). This is supported by the strong UVOC source inferred in this region (Fig. 6o) along with increased BVOC emissions in Canada and the Russian Far East (Fig. 6c). The latter effect was already present in the single-compound inversion and is mainly driven by TROPOMI HCHO columns (Section 4.1).

In the broadleaf and mixed forest zone, where monoterpenes are believed to contribute little to biogenic emissions (Kaiser 590 et al., 2015), the glyoxal underestimation appears to result primarily from missing knowledge of glyoxal production pathways from isoprene in the model: a strong UVOC source is inferred across the entire zone, while known biogenic emissions only increase locally, particularly in the western United States. This pattern suggests that the isoprene flux in the inventory is roughly correct in most areas, but the chemical pathways leading to glyoxal formation from isoprene are incompletely represented. This confirms the notion that current knowledge of the isoprene degradation mechanism at low-$NO_x$ remains incomplete (e. g., 595 Medeiros et al., 2022; Berndt et al., 2018). Indeed, the relatively low UVOC source over China, India, eastern Brazil, and to a lesser extent the eastern United States may indicate that glyoxal formation under high-$NO_x$ conditions is better captured in the model, whereas the low-$NO_x$ glyoxal formation yield might be strongly underestimated.

Although the top-down UVOC source is likely predominantly biogenic, a part of it may be misattributed due to the co-occurrence of biogenic emissions with biomass burning during hot and dry periods. In such conditions, elevated glyoxal 600 columns may result from both fire-related VOCs and enhanced BVOC emissions. The top-down UVOC source might then be overestimated due to a poor representation of secondary glyoxal formation from pyrogenic VOCs, as indicated by the





reported occurrence of high glyoxal in aged biomass burning plumes (Alvarado et al., 2020; Kluge et al., 2023) (Fig. 6n). The oxidation of furanoid compounds released by fires is believed to form glyoxal at significant yields (Romanias et al., 2024) and is currently ignored in large-scale models. A part of this glyoxal production might occur several days after emission, as it

involves long-lived intermediates such as maleic anhydride (Gkatzelis et al., 2024).

As pointed out in Kluge et al. (2023), a study based on aircraft data and the EMAC model, multiple sources might contribute to the model deficits. In particular, recent evidence that phospholipids during the death phase of algal blooms generate glyoxal upon oxidation (Williams et al., 2024) could partly explain the high UVOC source required at the tropical coastal regions (e.g., Atlantic coast of South America, Indonesia, Central America). Therefore, we acknowledge that the use of a single, continental

glyoxal precursor is overly simplified and that a multitude of glyoxal precursors and formation pathways are likely at play. In addition, it cannot be ruled out that the strong inferred UVOC source, especially in humid tropical regions, might be partly related to the high sensitivity of glyoxal retrievals to the water vapor cross-section (Lerot et al., 2021).

## 4.5   Evaluation of modeled formaldehyde columns against PGN data

The PGN measurement sites shown in Fig. 9 are listed with their coordinates in Table 8. The PGN HCHO columns (PGN,

2021) are compared with TROPOMI and model columns in Fig. 10. The 2021 averages of the observed and modeled columns are listed in Table 8. The left panel of Figure 10 displays a comparison of the bias-corrected (Eq. 1) TROPOMI and PGN HCHO columns for every measurement station and every month (where data was available) in 2021. The slope is almost equal to unity (1.03) but a slight offset is derived, with TROPOMI columns being on average circa $10^{15}$ molec. $\mathrm{cm}^{-2}$ lower than the PGN data. We note, however, that the TROPOMI columns are averaged over the $2° \times 2.5°$ grid cell in which the station is

located, whereas the PGN data represent localized measurements, often within an urban area. Furthermore, even though only PGN measurements taken within three hours of the TROPOMI overpass time were selected, the sampled days generally differ between TROPOMI and PGN. Taking these limitations into account, the good overall agreement between the ground-based and satellite data does seem to support the validity of the bias correction applied to TROPOMI columns based on FTIR data. The comparison between PGN and modeled columns (a priori and optimized) is shown in the right panel of Fig. 10. Here,

the modeled data are sampled at the PGN observation times. Similar to the comparison with TROPOMI, we find that the model columns are lower than PGN columns on average. The increased VOC emissions and HCHO columns after inversion (OPTHCHOGLY) result in a better match with the PGN data relative to the a priori, with the mean bias decreasing from approximately 30% to 20%.

At the large majority of locations (26 out of 35), the optimization improves the agreement between modeled and PGN

columns, as expressed by the average absolute deviations listed in Table 8. The average observed HCHO column across all stations listed in Table 8 is $10 \cdot 10^{15}$ molec. $\mathrm{cm}^{-2}$, while the averages from the a priori and optimized model are $7.3 \cdot 10^{15}$ molec. $\mathrm{cm}^{-2}$ and $7.7 \cdot 10^{15}$ molec. $\mathrm{cm}^{-2}$. At 5 locations (Bayonne, Bremen, Charles City, Cape Elizabeth, Chapel Hill and La Porte), the a priori model already closely matched the PGN data with an average difference less than $10^{15}$ molec. $\mathrm{cm}^{-2}$. At the Asian sites of Beijing, Busan and Seoul, the a priori model underprediction of approximately 30% is reduced to 20% after inversion.

An exception is the Incheon station, where the observed high column densities ($20 \cdot 10^{15}$ molec. $\mathrm{cm}^{-2}$) are poorly represented





**Table 8.** Observed HCHO columns ($10^{15}$ molec. cm$^{-2}$) from PGN stations, and corresponding values from a priori and optimized (OPTHCHOGLY) model. Ind.: index in Fig. 9; Country: ISO country code; Lat.: latitude ($^\circ$N); Long.: longitude ($^\circ$E); Obs.: observed column; A priori: a priori model; Optimized: optimized model; $\Delta_\mathrm{apr}$ and $\Delta_\mathrm{opt}$: average absolute deviation $\Delta$ calculated as the difference (HCHO$_\mathrm{model}$ − HCHO$_\mathrm{obs}$) for resp. the a priori and optimized model ($10^{15}$ molec. cm$^{-2}$). NCAR: National Center for Atmospheric Research; NIES: National Institute for Environmental Studies.

| Ind. | Site | Country | Lat. | Long. | Month | Obs. | A priori | Optimized | $\Delta_\mathrm{apr}$ | $\Delta_\mathrm{opt}$ |
|---|---|---|---|---|---|---|---|---|---|---|
| 1 | Altzomoni | MEX | 19.12 | 261.35 | 1-6, 12 | 6.9 | 2.5 | 3.1 | -4.4 | -3.8 |
| 2 | Athens | GRC | 37.99 | 23.77 | 1-12 | 14.3 | 5.4 | 5.7 | -8.9 | -8.6 |
| 3 | Bayonne | USA | 40.67 | 285.87 | 3-12 | 9.6 | 8.8 | 8.9 | -0.8 | -0.7 |
| 4 | Beijing | CHN | 40.01 | 116.38 | 7-12 | 12.1 | 8.2 | 9.4 | -3.9 | -2.7 |
| 5 | NCAR, Boulder | USA | 40.04 | 254.76 | 8-12 | 6.6 | 4.0 | 4.8 | -2.6 | -1.8 |
| 6 | Bremen | DEU | 53.08 | 8.81 | 5-11 | 7.4 | 6.7 | 6.4 | -0.7 | -1.0 |
| 7 | Bristol | USA | 40.11 | 285.12 | 3-12 | 9.7 | 8.6 | 8.8 | -1.1 | -0.9 |
| 8 | Busan | KOR | 35.24 | 129.08 | 3-12 | 12.5 | 8.7 | 9.6 | -3.8 | -2.9 |
| 9 | Cambridge | USA | 42.38 | 288.89 | 2-12 | 10.9 | 6.3 | 6.3 | -4.6 | -4.6 |
| 10 | Cape Elizabeth | USA | 43.56 | 289.79 | 6-8, 10-12 | 7.4 | 8.4 | 8.5 | 1.0 | 1.1 |
| 11 | Charles City | USA | 37.33 | 282.79 | 1-5 | 4.6 | 4.9 | 4.9 | 0.3 | 0.3 |
| 12 | Chapel Hill | USA | 35.97 | 280.91 | 3-4, 6-8 | 11.7 | 10.6 | 10.7 | -1.1 | -1.0 |
| 13 | Fairbanks | USA | 64.86 | 212.15 | 4-9 | 3.2 | 3.1 | 3.1 | -0.1 | -0.1 |
| 14 | Helsinki | FIN | 60.20 | 24.96 | 6-11 | 8.5 | 4.9 | 4.8 | -3.6 | -3.7 |
| 15 | Houston | USA | 29.72 | 264.66 | 7-12 | 16.1 | 10.0 | 10.5 | -6.1 | -5.6 |
| 16 | Incheon | KOR | 37.57 | 126.64 | 5-9 | 19.6 | 11.9 | 13.0 | -7.7 | -6.6 |
| 17 | La Porte | USA | 29.67 | 264.93 | 5-12 | 13.9 | 9.6 | 10.1 | -4.3 | -3.8 |
| 18 | Londonderry | USA | 42.86 | 288.62 | 4-12 | 6.7 | 7.1 | 7.1 | 0.4 | 0.4 |
| 19 | Manhattan, Kansas | USA | 39.10 | 263.39 | 3-7 | 10.8 | 7.4 | 7.8 | -3.4 | -3.0 |
| 20 | Mexico City | MEX | 19.33 | 260.82 | 2-12 | 14.8 | 8.1 | 10.3 | -6.7 | -4.5 |
| 21 | Manhattan, New York | USA | 40.81 | 286.05 | 3-5, 7-11 | 10.5 | 8.9 | 9.0 | -1.6 | -1.5 |
| 22 | Mountain View | USA | 37.42 | 237.94 | 1-12 | 8.0 | 6.5 | 7.8 | -1.5 | -0.2 |
| 23 | New Brunswick | USA | 40.46 | 285.57 | 1, 5-12 | 11.1 | 9.0 | 9.2 | -2.1 | -1.9 |
| 24 | New Haven | USA | 41.30 | 287.10 | 1-12 | 11.0 | 8.1 | 8.3 | -2.9 | -2.7 |
| 25 | Philadelphia | USA | 39.99 | 284.92 | 4-12 | 11.4 | 9.7 | 10.2 | -1.7 | -1.2 |
| 26 | Pittsburgh | USA | 40.47 | 280.04 | 9-12 | 8.3 | 6.0 | 6.0 | -2.3 | -2.3 |
| 27 | Seosan | KOR | 36.78 | 126.49 | 1-6, 10-12 | 7.1 | 6.0 | 6.5 | -1.1 | -0.6 |
| 28 | Seoul | KOR | 37.56 | 126.93 | 1-12 | 10.3 | 7.7 | 8.4 | -2.6 | -1.9 |
| 29 | Tel Aviv | ISR | 32.11 | 34.81 | 6-12 | 8.2 | 5.8 | 6.2 | -2.4 | -2.0 |
| 30 | NIES, Tsukuba | JPN | 36.05 | 140.12 | 8-12 | 9.2 | 6.4 | 6.6 | -2.8 | -2.6 |
| 31 | Tsukuba | JPN | 36.07 | 140.12 | 5-12 | 11.1 | 7.0 | 7.2 | -4.1 | -3.9 |
| 32 | Wallops Island | USA | 37.84 | 284.52 | 6-12 | 10.9 | 9.5 | 10.0 | -1.4 | -0.9 |
| 33 | Washington | USA | 38.92 | 282.99 | 1-12 | 11.7 | 7.7 | 8.0 | -4.0 | -3.7 |
| 34 | Wrightwood | USA | 34.38 | 242.32 | 1-8, 10-12 | 6.1 | 2.5 | 2.8 | -3.6 | -3.3 |
| 35 | Yokosuka | JPN | 35.32 | 139.65 | 1-12 | 9.2 | 8.2 | 8.4 | -1.0 | -0.8 |





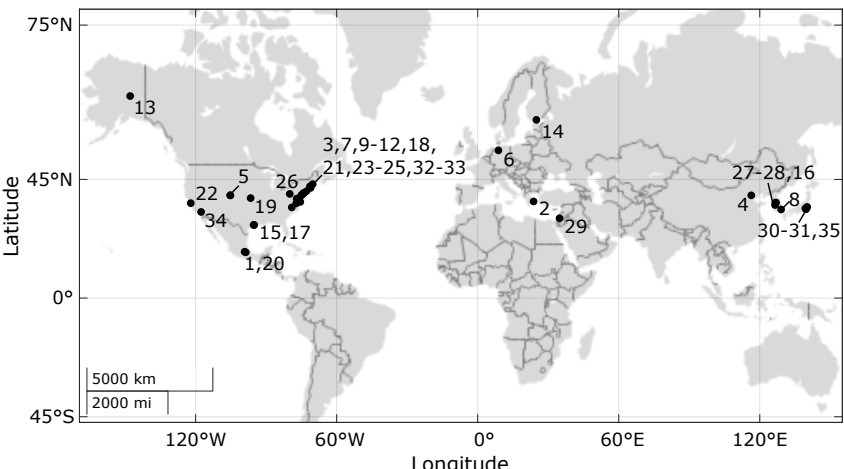

**Figure 9.** Location of PGN HCHO measurement sites used for independent model evaluation. Each index corresponds to a site in Table 8.

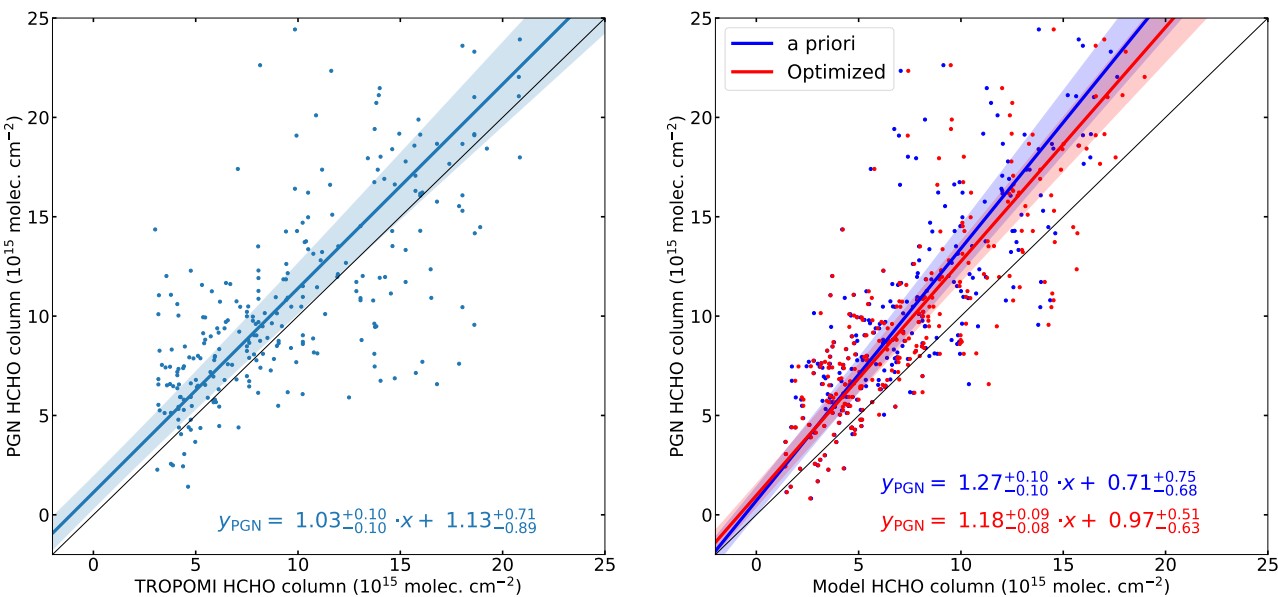

**Figure 10.** Scatter plot of monthly TROPOMI HCHO columns (left) and modeled (a priori and OPTHCHOGLY inversion) HCHO columns (right) versus ground-based PGN HCHO columns for 2021. The data have been spatially collocated and are monthly averaged. The linear TROPOMI fit is shown in light blue, and its 1-$\sigma$ uncertainties are depicted by the light blue-shaded area. Similarly, the a priori and optimized models (and their fit and uncertainties) are shown in blue and red, respectively.



in both the a priori ($12 \cdot 10^{15}$ molec. cm$^{-2}$) and the optimized model ($13 \cdot 10^{15}$ molec. cm$^{-2}$). Similarly for Tsukuba, located 50 km to the northwest of Tokyo, only a small improvement is found after optimization with the model remaining too low compared to the observation. At the European sites of Helsinki and Athens, the optimized model is unable to bridge the large model underestimation (43% and 60%, respectively). This poor performance is most likely due to the coarse resolution in this
study ($2° \times 2.5°$), since a high-resolution inversion (at $0.5° \times 0.5°$) over Europe constrained by TROPOMI HCHO columns derived a strong increase of isoprene emissions in Southern Europe in order to match the satellite observations (Oomen et al., 2024).

Across the 14 locations in the East Coast states of the U.S., the mean PGN column ($9.7 \cdot 10^{15}$ molec. cm$^{-2}$) is in good agreement with the a priori average ($8.1 \cdot 10^{15}$ molec. cm$^{-2}$) and the optimization brings about only small improvements
($8.3 \cdot 10^{15}$ molec. cm$^{-2}$, Table 8). At 9 of those sites (Bayonne, Bristol, Charles City, Chapel Hill, Londonderry, Manhattan New York, New Brunswick, Philadelphia, and Wallops Island) the a priori model already agreed well with the PGN data and continued to improve after optimization, while at Cambridge, New Haven, Pittsburgh, Washington, and Cape Elizabeth, the optimization results in minimal or no improvement. Elsewhere in the U.S., the a priori model underestimates observed columns by about 40% at Houston and Boulder, 60% at Wrightwood, 19% at Mountain View, and achieves an excellent
match at Fairbanks. The model performance after optimization is moderately improved at Houston (35%), Boulder (27%) and Wrightwood (54%), while the top-down model bias is close to zero at Mountain View. The PGN abundances at Altzomoni and Mexico City are much higher than the a priori model estimates and while the optimized abundances increase by about 25% after optimization, they remain much lower than the observed levels. Both of these locations are difficult to capture with the model, however, due to orographic effects (e.g., Altzomoni is located 4.2 km above sea level) and to intense urban emissions
causing strong heterogeneity of HCHO columns around Mexico City.

### 4.6   Evaluation of modeled glyoxal concentrations against in situ data

The comparisons between the model and in situ data were performed by linear interpolation of the gridded model values, while taking into account the month and hourly range of the measurements. Out of 25 rural sites (Fig. 11, Supplementary Table S1), the agreement between model and observations improves after optimization in 16 cases, as can be seen in Fig. 12. For most
campaigns carried out in temperate forests (Central Rocky Mountains, Pinnacles, Goldlauter, Pabstthum, Wangdu, Tomakomai, Moshiri and Cape Grim) the a priori glyoxal estimates already matched in situ observations reasonably well (with differences of less than 16 ppt), and the TROPOMI-constrained inversion further improves the agreement (to differences less than 8 ppt). The observations in the Sierra Nevada Mountains across different years are, on average, in relatively good agreement with the model (within 30%), but show a large variability (almost factor of 2). At the temperate site in the Yangtze River Delta, the high
observed glyoxal level stemmed from crop residue burning (Liu, J. et al., 2020) at the time of measurement, and can therefore not be replicated by the model for a different year (here 2021).

At the tropical rainforest sites, the in situ observations vary widely, resulting in a poor model match: in Manacapuru, a semi-rural city in the Amazon rainforest located 80 km away from the Amazonas capital Manaus, observed glyoxal levels are of the order of 0.01 ppb, significantly lower than the a priori estimate of 0.05 ppb, while in the pristine Borneo rainforest,





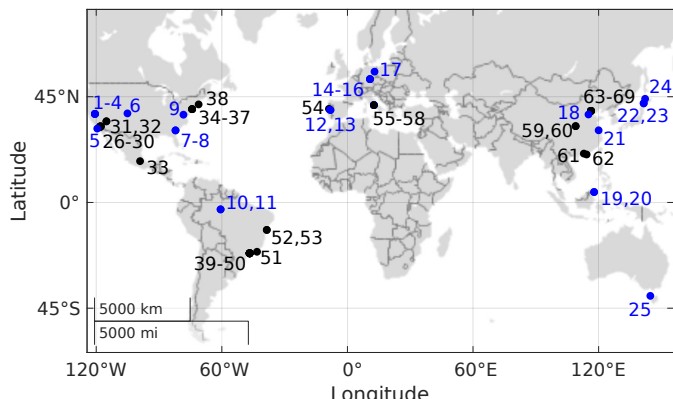

**Figure 11.** Location of in situ CHOCHO measurement sites (rural in blue, urban in black) used for model evaluation. Each index corresponds to (part of) a campaign in Fig. 12, Supplementary Fig. S4, and Supplementary Tables S1 and S2.

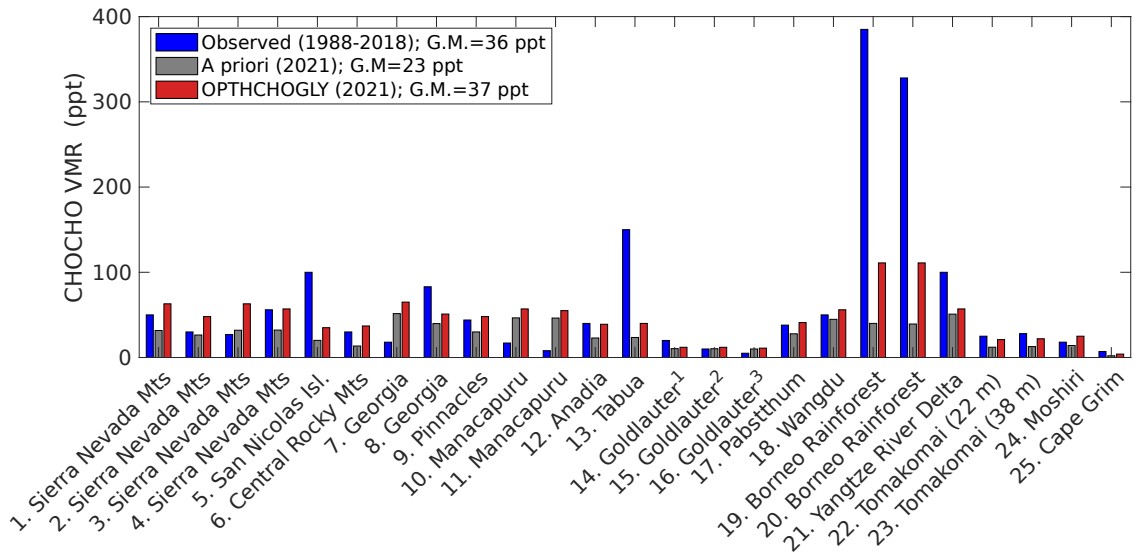

**Figure 12.** Observed CHOCHO mixing ratios (pptv) from in situ measurement campaigns at 17 rural sites (between 1990 and 2018), and corresponding values from a priori and optimized (OPTHCHOGLY) model. Numbering corresponds to the detailed entries in Supplementary Table S1 and the locations on the map in Fig. 11. G.M.: geometric mean. Different bars for the Sierra Nevada Mountains campaign correspond to measurements in different years. At Tomakomai, measurements were performed at 22 m and 38 m above ground level. Some subsets of the observations are listed for different times of the month: [1]late October, [2]early October, [3]mid-October.

the observed concentrations average almost 30 times higher, significantly higher than the a priori estimate. The TROPOMI-





constrained optimization results in an increased glyoxal concentration at all rainforest sites. In general, levels of glyoxal would be expected to be elevated in these areas due to the strong biogenic emissions of isoprene and monoterpenes.

In Tabua, Portugal, the high (150 ppt) observed glyoxal concentrations–as opposed to the moderate ones in Anadia–can be explained by the Tabua site's location, surrounded by large Eucalyptus globulus plantations. This species is known to strongly emit isoprene and monoterpenes, as confirmed by the measurements at the site (Cerqueira et al., 2003). Neither the a priori nor the OPTHCHOGLY model can resolve these elevated levels. In Georgia, U.S., the very low glyoxal levels in the July-August campaign cannot be replicated by the models. The nearly five times higher levels in the June campaign at the same site are closer to those resulting from the optimized model. Finally, at San Nicholas Island, the very high (100 ppt) glyoxal concentration measured was due to a severe smog event during the measurements (Grosjean et al., 1996) and is evidently not replicated by the 2021 model results.

Overall, across the rural sites, the mean glyoxal concentration inferred by the OPTHCHOGLY inversion (37 ppt) is in excellent agreement with the observations (36 ppt), and is on average 60% higher than the bottom-up model average (23 ppt). A very similar result is obtained when assuming a shorter UVOC lifetime (1 day, S1 sensitivity inversion), whereas the modeled average glyoxal concentration after inversion is circa 30% higher than the observation when assuming a long UVOC lifetime (10 days). At urban locations, the mean observed level (381 ppt) is about 10 times higher than typical rural values, and cannot be replicated by the inversions. As shown in Supplementary Table S2 and Fig. S4, the urban measurements strongly fluctuate with time and location, and therefore, as opposed to the rural measurements, have limited representativeness for the coarse model grid used in this work. For example, glyoxal levels in Salvador, Brazil (index 52 and 53 in Supplementary Table S2) were measured only one hour apart during the morning rush hour, but differ by more than a factor of 10.

## 4.7 Evaluation of modeled formaldehyde and glyoxal columns against long-term MAX-DOAS data at Phimai

At the rural site of Phimai, central Thailand (15.18°N, 102.56°E), continuous measurements of formaldehyde and glyoxal were obtained using the MAX-DOAS technique from October 2014 to October 2016 (Hoque et al., 2018) allowing for a comparison with the model throughout an entire year (Fig. 13). The climate in Phimai has two pronounced seasons: the dry season from January to April, and the wet season from June to September. Biomass burning is common during the dry season, and in combination with high biogenic emission fluxes, it leads to the enhanced formaldehyde (7 ppb) and glyoxal (0.2 ppb) levels in March, whereas the monthly mean mixing ratios during the wet season were 2 ppb and 0.1 ppb, respectively. The optimization improves the model performance at this site significantly, with the mean percentage bias between the model and the MAX-DOAS observation having decreased for both compounds, and the Pearson correlation coefficient having increased from 0.52 to 0.84 for formaldehyde and from 0.22 to 0.71 for glyoxal (Fig. 13a, c). The MAX-DOAS observations indicate dry season formaldehyde and glyoxal concentration levels about twice as high as in the wet season. While that ratio was close to unity in the inventory-based model (reflecting the low seasonal variability in Fig. 13a), the seasonality is more marked after optimization, in good agreement with the MAX-DOAS data. The comparison between TROPOMI and modeled data at the same location shows excellent top-down agreement, with a decrease of the mean percentage bias between the model and the TROPOMI observation after optimization from -20% to -5% for formaldehyde and from -21% to 0% for glyoxal (Fig. 13b, d).



The seasonal variation of the MAX-DOAS measurements is very well captured by the TROPOMI observations and reproduced
by the optimized (OPTHCHOGLY) model.

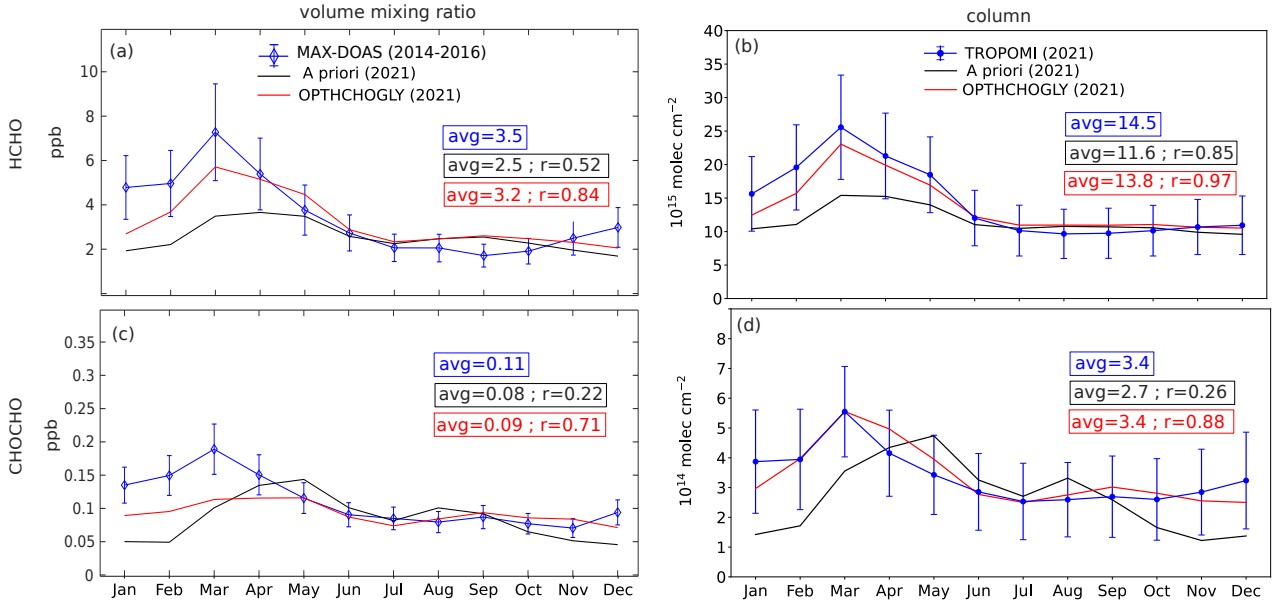

**Figure 13.** Monthly averaged formaldehyde (a, b) and glyoxal (c, d) levels at Phimai, Thailand. Panels (a, c): MAX-DOAS observations for
the 0–1 km layer obtained from October 2014 to October 2016 by Hoque et al. (2018) are shown in blue, the a priori and optimized model
for 2021 in black and red, respectively. Error bars represent the total observation error (including random, systematic and bias in elevation
angle) of 30% for formaldehyde and 20% for glyoxal. Panels (b, d): TROPOMI formaldehyde (b) and glyoxal (d) columns in 2021 are
shown in blue; error bars represent the total observation error (Section 3.2.1 and Section 3.2.2). The a priori and optimized (OPTHCHOGLY
inversion) formaldehyde and glyoxal columns for the site in 2021 are in black and red, respectively. Insets: avg = yearly average; r = Pearson
correlation coefficient between model and observation.

## 5 Conclusions

We performed a global top-down inversion of continental NMVOC emissions for 2021 constrained by glyoxal and formalde-
hyde columns from the spaceborne TROPOMI instrument. The inversion is realized using the adjoint of the MAGRITTEv1.2
model, which was updated to include detailed aromatic chemistry and up-to-date parameterization of glyoxal uptake onto
aerosols. This study represents the first joint inversion using both tracers from TROPOMI, allowing better constraints on VOC
sources than were possible with earlier, lower resolution satellite data. In addition, the joint inversion enables improved con-
straints on the longstanding issue of missing glyoxal sources in global atmospheric chemistry models. The total amount of
NMVOCs emitted from land into the atmosphere in 2021 is estimated in this study at 1070 Tg and composed of 749 Tg from




vegetation, 102 Tg from biomass burning (wildfires, land clearing, and agricultural and controlled burns), and 219 Tg from human activity (domestic, industrial, agricultural, energy production and transport). This estimate is by 19% higher than the bottom-up inventories used in this work. Two emission inversions were conducted. The first one, constrained by atmospheric formaldehyde alone, provides robust constraints on the total VOC flux (especially isoprene), because the yield of formaldehyde from the oxidation of major NMVOCs is relatively well known. The formaldehyde-and-glyoxal constrained inversion

allows the further partitioning of anthropogenic emissions into glyoxal and non-glyoxal precursors; furthermore, it addresses the quantification of the missing source of glyoxal.

The inversions also reveal significant regional and sectoral discrepancies between the bottom-up and top-down emissions. Relative to the MEGAN-based a priori inventory, isoprene emissions from the two-compound inversion are enhanced by as much as 90%, 58%, 42% and 28%, respectively, over Northern Hemisphere Africa, South Asia, Europe and North America.

Large increases (locally up to a factor of 3) are also inferred other regions, notably the western half of North America. Emission decreases are derived over Australia (by 18%) and large parts of South America. The results are broadly consistent with a recent inversion study based on (bias-corrected) OMI HCHO data, to the notable exception of middle and high latitudes.

Fire emissions in Southern Hemisphere Africa and South Asia are enhanced by about 50% relative to the QFED inventory, with India reaching 3 Tg yr$^{-1}$, three times the bottom-up value. Large mismatches over Zambia, Zimbabwe, and Mozambique

appear consistent with earlier studies. Other regions with large differences coincide with places where agricultural burning is common practice, showing that these remain poorly captured in inventories. In the northwestern United States, emissions from July 2021 wildfires are increased by nearly a factor of three, while Siberian fire emissions are well represented by the inventory.

Anthropogenic VOC emissions in the United States, Russia, and Europe remain mostly unaffected by the emission optimization. In contrast, large emission increases are derived over Iran (factor of ∼2) and over India (∼1.6), where optimized emissions

reach 19 Tg in 2021. After inversion, global emissions of anthropogenic glyoxal precursors (acetylene, ethene, and aromatic hydrocarbons) are estimated at 50 Tg yr$^{-1}$, 43% higher than the bottom-up total from the CAMS-GLOB-ANT inventory. Global acetylene (4.4 Tg yr$^{-1}$) and ethene (10.2 Tg yr$^{-1}$) emissions are both about 60% higher than in the a priori, whereas benzene emissions (7.9 Tg yr$^{-1}$) are ∼40% higher. India is identified as the top global emitter of acetylene and ethene, while also emitting large amounts of aromatics. Although recently introduced air quality regulations may improve activity reporting

in India, reliable emission estimation also requires the development of emission factors tailored to the Indian context.

China emerges as the largest emitter of aromatic hydrocarbons: 6.5 Tg in 2021, out of which 1.8 Tg in Beijing-Tianjin-Hebei alone, ∼30% more than reported in CAMS-GLOB-ANT. Anthropogenic emissions from Mexico, the Middle East, Indonesia, and parts of Africa are also substantially enhanced; for example, inferred aromatic emissions from Africa are 1.6 times higher than inventory values (4.7 Tg yr$^{-1}$ vs 2.9 Tg yr$^{-1}$). Given the health concerns of exposure to aromatics, for example the

estimate by Partha et al. (2022) that more than half a million preterm births per year are attributable to aromatic hydrocarbon exposure in China and India, it is important to improve the accuracy of these emission estimates.

Based on the joint inversion of TROPOMI formaldehyde and glyoxal columns, we estimate that 41% of the global glyoxal source cannot be explained by currently known VOC emissions and chemical mechanisms. The inversion yields a total glyoxal source of ca. 100 Tg yr$^{-1}$, of which 58 Tg is attributed to the photochemical production from known VOC sources, and 41 Tg



from unidentified VOCs (UVOCs). The missing source appears predominantly biogenic, with 70% originating in the Tropics. Given the dominance of isoprene in the global VOC budget, a large part of the missing glyoxal source over broadleaved and mixed forests likely reflects underrepresented or missing chemical pathways of glyoxal formation from isoprene, especially under low-$NO_x$ conditions. The low magnitude of the inferred missing source over China, India and the eastern U.S. suggests that glyoxal formation from isoprene is well represented in the model at higher $NO_x$ levels. In the boreal zone, both an underestimation of monoterpene emissions in the MEGAN inventory and an underestimation of glyoxal yields from monoterpenes in the model chemistry might additionally contribute to the discrepancy. Beyond the results of this study, other potential contributors to the glyoxal budget have been suggested. Secondary production from furanoids during fire events may play a role (Romanias et al., 2024), pointing to the need for their future inclusion in chemical transport models. Similarly, marine glyoxal sources could be relevant in coastal regions, although the contribution of algae as precursors remains poorly understood (Williams et al., 2024).

The modeled formaldehyde and glyoxal columns after inversion show good agreement in both magnitude and seasonality with the TROPOMI observations, with very few exceptions. Evaluation against independent datasets confirms the robustness of the inversion. More specifically, comparison with Pandonia Global Network column data further validates the (bias-corrected) HCHO TROPOMI dataset and shows a substantial reduction of the mean model bias (from 30% to 20%) and improved statistics at the majority of stations. In addition, model comparison with in situ glyoxal concentrations shows significantly improved agreement at rural stations: the mean glyoxal concentration after the inversion closely matches the observations, whereas the bottom-up model average was ∼40% too low. Finally, comparison with a one-year MAX-DOAS dataset at a tropical rural site in Thailand also shows substantial improvements in both absolute concentrations and seasonality, for formaldehyde as well as for glyoxal, giving credence to the emission updates in this region strongly influenced by biogenic precursors.

*Data availability.* The global top-down VOC emission fluxes for five categories, constrained by TROPOMI formaldehyde and glyoxal observations, are available at https://doi.org/10.18758/52E4U9EN (last accessed: 20 August 2025). (Sfendla et al., 2025). The Copernicus Sentinel-5P TROPOMI Level 2 Formaldehyde Total Column products (v2) can be found at https://doi.org/10.5270/S5P-vg1i7t0, and the TROPOMI glyoxal tropospheric columns (v4) at https://doi.org/10.18758/71021069. The MEGAN-MOHYCAN isoprene inventory is available at https://emissions.aeronomie.be.

*Author contributions.* YS carried out the analysis, performed the evaluation of the model against in situ data, and wrote the manuscript. TS and JFM designed the inversions, performed the optimizations, and reviewed the manuscript. GMO performed the evaluation of the model against the PGN column data. BO contributed to the a priori model configuration. IDS provided the TROPOMI formaldehyde retrievals and advice on their use. TD developed the glyoxal retrievals from TROPOMI and gave advice on their use. All the co-authors read and commented on the manuscript and provided feedback.



*Competing interests.*   The contact author has declared that none of the authors has any competing interests.

*Acknowledgements.*   YS was supported by the GLANCE project funded by the European Space Agency (ESA) under the "eo science for society" programme and by the CONCERTO project funded by the European Commission under the Horizon Europe programme (grant agreement no. 101185000). GMO was supported by Belspo through the ProDEx project TROVA-E3 (TROPOMI Validation - Phase E3) of the European Space Agency (2024-2025).



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
