# Peer review of "Global VOC emissions quantified from inversion of TROPOMI spaceborne formaldehyde and glyoxal data"

_EGUsphere, 2025_

## Author Comment (AC1)

**Reply to the comments of Reviewer#1**

We sincerely thank the reviewer for the positive evaluation of our manuscript and the constructive comments. The reviewer's comments are in black, our responses in blue.

This manuscript presents an important study that uses TROPOMI HCHO and CHOCHO data to quantify global VOC emissions from biogenic, pyrogenic and anthropogenic sources. The results indicate a large proportion of unidentified VOCs over the tropics, which is an important finding that will motivate a lot of future studies. The manuscript is very well done, and the results are interesting and convincing. I only have a few minor comments.

1. The inversion framework relies on TROPOMI HCHO and CHOCHO retrievals, but these products are themselves subject to uncertainties. As the authors show, CTMs substantially underestimate CHOCHO, yet the TROPOMI retrievals use CTM a priori profiles to compute air mass factors. It is unclear how uncertainties in these a priori profiles propagate into the inversion. It would be informative to assess how the retrieved HCHO and CHOCHO column densities would change if air mass factors were recalculated using CTM fields updated with the optimized VOC emissions.

The reviewer is right to point out that the vertical profiles assumed in TROPOMI HCHO and CHOCHO retrievals are a large source of uncertainty. However, the error caused by vertical profile shape errors (the so-called smoothing error) is taken care of through the application of the satellite averaging kernels to the model profiles (see for example, Lerot et al., 2021). This process removes errors due to vertical profile inconsistencies in the comparison of satellite columns with "smoothed" model columns. We changed the last sentence of Sect. 3.1 as follows: "These kernels are applied to the modeled vertical profiles to account for the instrument's altitude-dependent sensitivity and remove errors due to vertical profile inconsistencies in the comparison of satellite columns with the model (Oomen et al., 2024)."

2. The inversion framework appears to attribute model-satellite discrepancies in HCHO and CHOCHO solely to emission errors. However, both species are secondary products, and their yields depend on chemical mechanisms and NOx levels. It is therefore not clear how much of the discrepancy arises from uncertainties in HCHO and CHOCHO production pathways rather than emission errors. It would be helpful for the authors to comment on how the inversion accounts for, or is affected by, these chemical uncertainties.

Agreed. We added the following text discussing areas of uncertainty in the derivation of top-down emissions (Sect. 3.2). "The inverted emissions have uncertainties due to several factors affecting the HCHO columns, besides the magnitude of the emissions, such as the background HCHO levels due to methane oxidation, incomplete or incorrect information regarding VOC speciation in emission inventories, the VOC oxidation

mechanisms, the deposition of oxidation intermediates, the transport processes influencing the vertical profile of chemical compounds, and the NOx concentrations, known to influence the OH levels as well as the yields of HCHO and CHOCHO from key VOCs including isoprene."

3. Because satellite retrievals are available only under clear-sky conditions, sampling biases in HCHO and CHOCHO are likely. The authors note that CHOCHO sinks may differ under cloudy conditions, but it is unclear how such sampling biases are treated within the inversion framework and how they might influence the emission estimates. A brief discussion of this issue would strengthen the manuscript.

Indeed, the abundances of both HCHO and CHOCHO are affected by cloudiness. As explained in the manuscript, only observations with a cloud fraction less than 20% for CHOCHO and less than 40% for HCHO were retained for processing. However, we compare the satellite monthly averages to corresponding MAGRITTE monthly averages calculated from daily values accounting for the number of measurements (and averaging kernels) for each day. This has been now clarified in the manuscript. The last paragraph of Sect. 3.1 now reads "(...) the modelled monthly averaged columns are based on daily values at the satellite overpass time (~13:30 local time), while accounting for the number of observations and averaging kernels provided with the TROPOMI retrievals. These kernels..."

4. TROPOMI overpasses occur around 2 PM local time, when biogenic VOC emissions typically peak. It is not clear whether the inferred biogenic VOC emissions represent instantaneous emissions at overpass time or whether they are scaled to a daily mean. Clarification on this point would help interpret the emission magnitudes.

No, the top-down biogenic VOC emissions do not represent instantaneous emissions at the overpass time. The temporal variability of emissions is kept identical in the optimised and prior models (see Equation 4 in the manuscript).

5. Figure 4: Add figure legend.

Done as requested.

References

Lerot, C. et al., Glyoxal tropospheric column retrievals from TROPOMI - multi-satellite intercomparison and ground-based validation, Atmos. Meas. Tech., 14, 7775-7807, https://doi.org/10.5194/amt-14-7775-2021, 2021.

Oomen, G.-M. et al., Weekly-derived top-down volatile-organic-compound fluxes over Europe from TROPOMI HCHO data from 2018 to 2021, Atmos. Chem. Phys., 24, 449-474, https://doi.org/10.5194/acp-24-449-2024, 2024.

---

## Author Comment (AC2)

**Reply to the comments of Reviewer#2**

We sincerely thank the reviewer for the positive evaluation of our manuscript and the constructive comments. The reviewer's comments are in black, our responses in blue.

This work developed posteriori emissions of VOCs (Volatile Organic Compounds) by sectors based on the first joint inversion of TROPOMI formaldehyde and glyoxal columns using the adjoint of the MAGRITTE model. This is very important work given the high uncertainties of VOC emissions at both regional and global scales. The methodology is solid, the figures are great, and the analyses are comprehensive. I recommend authors to provide more clarifications on the chemical characteristics of the unidentified VOCs. Apart from the bias in the total VOC emissions, the speciation process is another source of uncertainty and can contribute to the bias of formaldehyde and glyoxal simulations. It would be great if this can be discussed in the paper. My detailed comments are provided as below.

Page 1, Line 10: for the unidentified VOCs, how does the model deal with this species in chemistry?

Thank you for this question. As explained in the manuscript, the photochemical oxidation of the unidentified UVOC precursors (UVOC) forms glyoxal with a molar yield of unity, resulting in a 5-day assumed lifetime. We've added the following sentence: "UVOC is assumed to react with OH at a rate constant equal to $2.315 \times 10^{15}$ cm$^3$ molec.$^{-1}$ s$^{-1}$, resulting in a lifetime of 5 days for [OH] = $10^6$ molec. cm$^{-3}$."

Page 5, Line 130-133: it's really nice to see the uncertainty analyses for the HCHO and CHOCHO retrievals from TROPOMI.

Thank you very much for your appreciation of the analysis.

Page 9, Line 266: the uptake coefficient is really high. Is it the initial uptake coefficient or the coefficient at stable?

Yes, this published value ($2.9 \times 10^{-3}$) is quite high (Liggio et al., 2005). It is the reactive uptake coefficient, not the initial uptake (or accommodation) coefficient. The text has been modified to reflect this: "(…) with a uniform reactive uptake probability..."

Page 10, Sect. 3.2: in the inversion, how does it work to provide emission constraints for different glyoxal precursors? Does it keep the relative percentage (VOC profiles) unchanged, and only tune the total emissions, or the profiles are also tuned?

As seen on Equation 4, the control parameters ($f$ vector) are defined per emission category. In other words, within each emission category (defined in Table 3), the multiplying factors ($\exp(f_j)$) are identical for all species included in the category, i.e. all species are changed in the same proportion. We added the following sentence in Sect.

3.2: "Note that the emission parameters are defined per category, i.e. the speciation within each category is unchanged by the inversion."

Page 17, Fig. 3: different revision directions for Eastern US and Eastern China based on OMI and TROPOMI. Can you elaborate more on this?

The reasons for these differences are likely related to differences in the HCHO columns between OMI and TROPOMI. Comparison of Figure 1a,b of our manuscript with Figure 6a,d of Müller et al. (2024) shows that the OMI (bias-corrected) HCHO columns are generally higher over Eastern China and Eastern US, compared to TROPOMI. We inserted the following text in the manuscript: "The reasons for these differences are likely related to differences in the HCHO columns between OMI and TROPOMI, especially at mid-latitudes. Indeed, the (bias-corrected) HCHO columns from TROPOMI (Fig. 1a,b) are generally lower than the corresponding OMI columns used by Muller et al. (2024) (their Fig. 6a,d) at mid-latitudes, and more specifically over Eastern US, Eastern China and northern Europe."

Page 19, Line 445: still curious about the chemistry of UVOC.

See above.

Page 20, Fig. 4: can you add legends in the figure? Although you have described this in the caption, it would be better to show them directly in the figure too.

Done as requested.

Page 22, Line 476: this is very useful information. The simulations over Southeast US are always off, so maybe we need both HCHO and CHOCHO constraints to revise the biogenic emissions.

As seen on Fig. 3, the biogenic emissions show a decrease after optimisation over the southeast US, when only TROPOMI HCHO is used as constraint. Fig. 6 shows essentially the same patterns, but with somewhat higher emissions.

Page 22, Line 497: Apart from the total VOCs, the speciation can play an important role in model simulations. I understand it's not quantified in the inversion, but can you explain more about the potential role of VOC speciation in your analyses, especially glyoxal inversion?

The referee is correct that the inversion results are dependent on the speciation of VOCs in the inventories. We added the following text discussing areas of uncertainty in the derivation of top-down emissions (Sect. 3.2). "The inverted emissions have uncertainties due to several factors affecting the HCHO columns, besides the magnitude of the emissions, such as the background HCHO levels due to methane oxidation, incomplete or incorrect information regarding VOC  speciation in emission inventories, the VOC oxidation mechanisms, the deposition of oxidation intermediates, the transport

processes influencing the vertical profile of chemical compounds, and the NOx concentrations, known to influence the OH levels as well as the yields of HCHO and CHOCHO from key VOCs including isoprene."

Page 24, Line 521-523: the same question as #4. I'm curious how the inversion model derives the optimized emissions for glyoxal precursors.

See above.

Sect. 4.5 - Sect. 4.7 are all about the model evaluations. Can you re-organize these sections? It would be better move them from Results to a new section like "Model evaluations".

Done as requested.

References

Liggio et al., Reactive uptake of glyoxal by particulate matter, J. Geophys. Res.-Atmos., 110, 1-13, https://doi.org/10.1029/2004JD005113, 2005.

Müller, J.-F., et al., Bias correction of OMI HCHO columns based on FTIR and aircraft measurements and impact on top-down emission estimates, Atmos. Chem. Phys., 24, 2207-2237, https://doi.org/10.5194/acp-24-2207-2024, 2024.